# Multifactor Sequential Disentanglement via Structured Koopman Autoencoders

**Nimrod Berman**[*]**, Ilan Naiman**[*]**, Omri Azencot**
Department of Computer Science
Ben-Gurion University of the Negev
{bermann,naimani}@post.bgu.ac.il, azencot@cs.bgu.ac.il

## Abstract

Disentangling complex data to its latent factors of variation is a fundamental task in representation learning. Existing work on sequential disentanglement mostly provides two factor representations, i.e., it separates the data to time-varying and time-invariant factors. In contrast, we consider *multifactor* disentanglement in which multiple (more than two) semantic disentangled components are generated. Key to our approach is a strong inductive bias where we assume that the underlying dynamics can be represented linearly in the latent space. Under this assumption, it becomes natural to exploit the recently introduced Koopman autoencoder models. However, disentangled representations are not guaranteed in Koopman approaches, and thus we propose a novel spectral loss term which leads to structured Koopman matrices and disentanglement. Overall, we propose a simple and easy to code new deep model that is fully unsupervised and it supports multifactor disentanglement. We showcase new disentangling abilities such as swapping of individual static factors between characters, and an incremental swap of disentangled factors from the source to the target. Moreover, we evaluate our method extensively on two factor standard benchmark tasks where we significantly improve over competing unsupervised approaches, and we perform competitively in comparison to weakly- and self-supervised state-of-the-art approaches. The code is available at GitHub.

## 1 Introduction

Representation learning deals with the study of encoding complex and typically high-dimensional data in a meaningful way for various downstream tasks (Goodfellow et al., 2016). Deciding whether a certain representation is better than others is often task- and domain-dependent. However, disentangling data to its underlying explanatory factors is viewed by many as a fundamental challenge in representation learning that may lead to preferred encodings (Bengio et al., 2013). Recently, several works considered two factor disentanglement of sequential data in which time-varying features and time-invariant features are encoded in two separate sub-spaces. In this work, we contribute to the latter line of work by proposing a simple and efficient unsupervised deep learning model that performs *multifactor* disentanglement of sequential data. Namely, our method disentangles sequential data to more than two semantic components.

One of the main challenges in disentanglement learning is the limited access to labeled samples, particularly in real-world scenarios. Thus, prior work on sequential disentanglement focused on unsupervised models which uncover the time-varying and time-invariant features with no available labels (Hsu et al., 2017; Li & Mandt, 2018). Specifically, two feature vectors are produced, representing the dynamic and static components in the data, e.g., the motion of a character and its identity, respectively. Subsequent works introduce two factor self-supervised models which incorporate supervisory signals and a mutual information loss (Zhu et al., 2020) or data augmentation and a contrastive penalty (Bai et al., 2021), and thus improve the disentanglement abilities of prior baseline models. Yamada et al. (2020) proposed a probabilistic model with a ladder module, allowing certain multifactor disentanglement capabilities. Still, to the best of our knowledge, the majority of existing work do not explore the problem of unsupervised multifactor sequential disentanglement.

---

[*]joint first authors

In the case of static images, multiple disentanglement approaches have been proposed (Kulkarni et al., 2015; Higgins et al., 2017; Kim & Mnih, 2018; Chen et al., 2018; 2016; Burgess et al., 2018; Kumar et al., 2017; Bouchacourt et al., 2018). In addition, there are several approaches that support disentanglement of the image to multiple distinct factors. For instance, Li et al. (2020) design an architecture which learns the shape, pose, texture and background of natural images, allowing to generate new images based on combinations of disentangled factors. In (Xiang et al., 2021), the authors introduce a weakly-supervised framework where $N$ factors can be disentangled, given $N-1$ labels. In comparison, our approach is fully unsupervised, deals with sequential data and the number of distinct components is determined by a hyperparameter.

Recently, Locatello et al. (2019) showed that unsupervised disentanglement is impossible without inductive biases on models and datasets. While exploiting the underlying temporal structure had been shown as a strong inductive bias in existing disentanglement approaches, we argue in this work that a stronger assumption should be considered. Specifically, based on Koopman theory (Koopman, 1931) and practice (Budišić et al., 2012; Brunton et al., 2021), we assume that there exists a learnable representation where the dynamics of input sequences becomes *linear*. Namely, the temporal change between subsequent latent feature vectors can be encoded with a matrix that approximates the *Koopman operator*. Indeed, the same assumption was shown to be effective in challenging scenarios such as fluid flows (Rowley et al., 2009) as well as other application domains (Rustamov et al., 2013; Kutz et al., 2016). However, it has been barely explored in the context of disentangled representations.

In this paper, we design an autoencoder network (Hinton & Zemel, 1993) that is similar to previous Koopman methods (Takeishi et al., 2017; Morton et al., 2018), and which facilitates the learning of linear temporal representations. However, while the dynamics is encoded in a Koopman operator, disentanglement is *not* guaranteed. To promote disentanglement, we make the following key observation: eigenvectors of the approximate Koopman operator represent time-invariant and time-variant factors. Motivated by this understanding, we propose a novel spectral penalty term which splits the operator's spectrum to separate and clearly-defined sets of static and dynamic eigenvectors. Importantly, our framework naturally supports multifactor disentanglement: every eigenvector represents a unique disentangled factor, and it is considered static or dynamic based on its eigenvalue.

**Contributions.** Our main contributions can be summarized as follows.

1. We introduce a strong inductive bias for disentanglement tasks, namely, the dynamics of input sequences can be encapsulated in a matrix. This assumption is backed by the rich Koopman theory and practice.

2. We propose a new unsupervised Koopman autoencoder learning model with a novel spectral penalty on the eigenvalues of the Koopman operator. Our approach allows straightforward multifactor disentanglement via the eigendecomposition of the Koopman operator.

3. We extensively evaluate our method on new multifactor disentanglement tasks, and on several two factor benchmark tasks, and we compare our work to state-of-the-art unsupervised and weakly-supervised techniques. The results show that our approach outperforms baseline methods in various quantitative metrics and computational resources aspects.

## 2 RELATED WORK

**Sequential Disentanglement.** Most existing work on sequential disentanglement is based on the dynamical variational autoencoder (VAE) architecture (Girin et al., 2020). Initial attempts focused on probabilistic models that separate between static and dynamic factors, where in (Hsu et al., 2017) the joint distribution is conditioned on the mean, and in (Li & Mandt, 2018) conditioning is defined on past features. Subsequent works proposed self-supervised approaches that depend on auxiliary tasks and supervisory signals (Zhu et al., 2020), or on additional data and contrastive penalty terms (Bai et al., 2021). In Han et al. (2021a), the authors replace the common Kullback–Leibler divergence with the Wasserstein distance between distributions. Some approaches tailored to video disentanglement use generative adversarial network (GAN) architectures (Villegas et al., 2017; Tulyakov et al., 2018) and a recurrent model with adversarial loss (Denton & Birodkar, 2017). Finally, Yamada et al. (2020) proposed a variational autoencoder model including a ladder module (Zhao et al., 2017), which allows to disentangle multiple factors. The authors demonstrated qualitative results of multifactor latent traversal between various two static features and three dynamic features on the Sprites dataset.

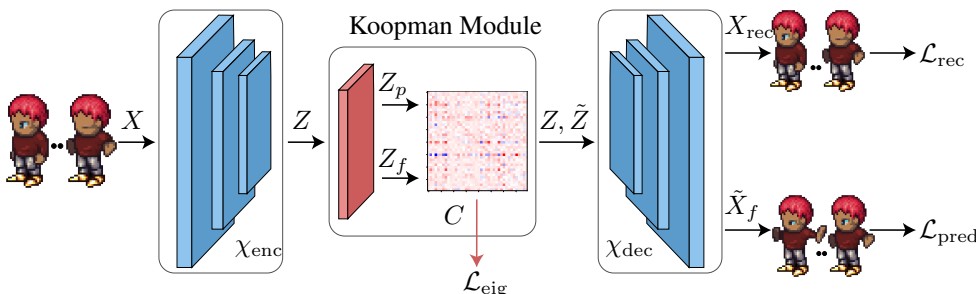

Figure 1: Our architecture is based on a Koopman autoencoder network which includes encoder $\chi_{\text{enc}}$, decoder $\chi_{\text{dec}}$, and a Koopman module that computes the Koopman operator $C$ via least squares solves. We augment this model with a novel spectral penalty term $\mathcal{L}_{\text{eig}}$ which facilitates the learning of spectrally structured $C$ matrices, and thus supporting multifactor disentanglement by construction.

**Dynamics Learning.** Over the past few years, an increasing interest was geared towards learning and representing dynamical systems using deep learning techniques. Two factor disentanglement methods based on Kalman filter (Fraccaro et al., 2017), and state-space models (Miladinović et al., 2019) focus on ordinary differential equation systems. Other methods utilize the mutual information between past and future to estimate predictive information Clark et al. (2019); Bai et al. (2020). Mostly related to our approach are Koopman autoencoders (Lusch et al., 2018; Yeung et al., 2019; Otto & Rowley, 2019; Li et al., 2019; Azencot et al., 2020; Han et al., 2021b), related to classical learning methods, e.g., Azencot et al. (2019); Cohen et al. (2021). Specifically, in (Takeishi et al., 2017; Morton et al., 2018; Iwata & Kawahara, 2020) the Koopman operator is learned via a least squares solve per batch, allowing to train a single neural model on multiple initial conditions. We base our architecture on the latter works, and we augment it with a novel spectral loss term which promotes disentanglement. Recently, an intricate model for video disentanglement was proposed in (Comas et al., 2021). While the authors employ Koopman techniques in that work, it is only partially related to our work since they explicitly model pose and appearance components, whereas our approach can model an arbitrary number of disentangled factors. In addition, their architecture is based on the attention network (Bahdanau et al., 2014), where the Koopman module is mostly related to prediction. In comparison, in our work the Koopman module is directly responsible for unsupervised disentanglement of sequential data.

**Koopman Spectral Analysis.** Our method is based on learning Koopman operators with structured spectra. Spectral analysis of Koopman operators is an active research topic (Mezić, 2013; Arbabi & Mezic, 2017; Mezic, 2017; Das & Giannakis, 2019; Naiman & Azencot, 2023). We explore Koopman eigenfunctions associated with the eigenvalue 1. These eigenfunctions are related to global stability (Mauroy & Mezić, 2016), and to orbits of the system (Mauroy & Mezić, 2013; Azencot et al., 2013; 2014). Other attempts focused on computing eigenfunctions for a known spectrum (Mohr & Mezić, 2014). Recently, pruning weights of neural networks using eigenfunctions with eigenvalue 1 was introduced in (Redman et al., 2021). However, to the best of our knowledge, our work is among a few to propose a deep learning model for generating spectrally-structured Koopman operators.

## 3 KOOPMAN AUTOENCODER MODELS

We recall the Koopman autoencoder (KAE) architecture introduced in (Takeishi et al., 2017) as it is the basis of our model. The KAE model consists of an encoder and decoder modules, similarly to standard autoencoders, and in between, there is a Koopman module. The general idea behind this architecture is that the encoder and decoder are responsible to generate effective representations and their reconstructions, driven by the Koopman layer which penalizes for nonlinear encodings.

We denote by $X \in \mathbb{R}^{b \times (t+1) \times m}$ a batch of sequence data $\{x_{ij}\} \subset \mathbb{R}^m$ where $i \in \{1, \ldots, b\}$ and $j \in \{1, \ldots, t+1\}$ represent the batch sample and time indices, respectively. The tensor $X$ is encoded to its latent representation $Z \in \mathbb{R}^{b \times (t+1) \times k}$ via $Z = \chi_{\text{enc}}(X)$. The Koopman layer splits the latent variables to past $Z_p$ and future $Z_f$ observations, and then, it finds the best linear map $C$ such that

$Z_p \cdot C \approx Z_f$. Formally, $Z_p = (z_{ij}) \in \mathbb{R}^{b \cdot t \times k}$ for $j \in \{1, \ldots, t\}$ and any $i$, and $Z_f = (z_{ij}) \in \mathbb{R}^{b \cdot t \times k}$ for $j \in \{2, \ldots, t + 1\}$ and any $i$, i.e., $Z_p$ holds the first $t$ latent variables per sample, and $Z_f$ holds the last $t$ variables. Then, $C = \arg\min_{\tilde{C}} |Z_p \cdot \tilde{C} - Z_f|_F^2 = Z_p^+ Z_f$, where $A^+$ denotes the pseudo-inverse of the matrix $A$. Importantly, the matrix $C$ is computed per $Z$ during both training and inference, and in particular, $C$ is not parameterized by network weights. Additionally, the pseudo-inverse computation supports backpropagation, and thus it can be used during training (Ionescu et al., 2015). Lastly, the latent samples are reconstructed with the decoder $X_{\text{rec}} = \chi_{\text{dec}}(Z)$.

The above architecture employs reconstruction and prediction loss terms: the reconstruction loss promotes an autoencoder learning, and the prediction loss aims to capture the dynamics in $C$. We use the notation $\mathcal{L}_{\text{MSE}}(X, Y) = \frac{1}{b \cdot t} \sum_{i,j} |Y(i,j) - X(i,j)|_2^2$ for the average distance between tensors $X, Y \in \mathbb{R}^{b \times t \times k}$ for $i \in \{1, \ldots, b\}$ and $j \in \{1, \ldots, t\}$. Then, the losses are given by

$$\mathcal{L}_{\text{rec}}(X_{\text{rec}}, X) = \mathcal{L}_{\text{MSE}}(X_{\text{rec}}, X) \,, \tag{1}$$

$$\mathcal{L}_{\text{pred}}(\tilde{Z}_f, Z_f, \tilde{X}_f, X_f) = \mathcal{L}_{\text{MSE}}(\tilde{Z}_f, Z_f) + \mathcal{L}_{\text{MSE}}(\tilde{X}_f, X_f) \,, \tag{2}$$

where $\tilde{Z}_f := Z_p \cdot C$, $\tilde{X}_f := \chi_{\text{dec}}(\tilde{Z}_f)$, and $X_f$ are the inputs corresponding to $Z_f$ latent variables. The network loss is taken to be $\mathcal{L} = \lambda_{\text{rec}} \mathcal{L}_{\text{rec}} + \lambda_{\text{pred}} \mathcal{L}_{\text{pred}}$, where $\lambda_{\text{rec}}, \lambda_{\text{pred}} \in \mathbb{R}^+$ balance between the reconstruction and prediction contributions. We show in Fig. 1 an illustration of the Koopman autoencoder architecture using the notations above.

## 4  MULTIFACTOR DISENTANGLING KOOPMAN AUTOENCODERS

How disentanglement can be achieved given the Koopman autoencoder architecture? For comparison, other disentanglement approaches typically represent the disentangled factors explicitly. In contrast the batch dynamics in KAE models is encoded in the approximate Koopman operator matrix $C$, where $C$ propagates latent variables through time while carrying the static as well as dynamic information. Thus, the time-varying and time-invariant factors are still entangled in the Koopman matrix. We now show that KAE theoretically enables disentanglement under the following analysis.

**Koopman disentanglement.** In general, one of the key advantages of Koopman theory and practice is the linearity of the Koopman operator, allowing to exploit tools from linear analysis. Specifically, our approach depends heavily on the spectral analysis of the Koopman operator (Mezić, 2005). In what follows, we perform our analysis directly on $C$, and we refer the reader to App. A and the references therein for a detailed treatment of the full Koopman operator. The eigendecomposition of $C$ consists of a set of left eigenvectors $\{\phi_i \in \mathbb{C}^k\}$ and a set of eigenvalues $\{\lambda_i \in \mathbb{C}\}$ such that

$$\phi_i^T C = \lambda_i \phi_i^T \,, \quad i = 1, \ldots, k \,. \tag{3}$$

The eigenvectors can be viewed as approximate Koopman eigenfunctions, and thus the eigenvectors hold fundamental information related to the underlying dynamics. For instance, the eigenvectors describe the temporal change in latent variables. Formally,

$$z_j^T C = \sum_{i=1}^k \langle z_j^T, \phi_i^T \rangle \phi_i^T C = \sum_i \bar{z}_j^i \lambda_i \phi_i^T \approx z_{j+1}^T \,, \quad j = 1, \ldots, t \,, \tag{4}$$

where $\bar{z}_j^i := \langle z_j^T, \phi_i^T \rangle$ is the projection of $z_j^T$ on the eigenvector $\phi_i^T$. The approximation follows from $C$ being the best (and not necessarily exact) linear fit between past and future features. Moreover, it follows that predicting step $j + r$ from $j$ is achieved simply by applying powers of the Koopman matrix on $z_j^T$, i.e., $z_j^T C^r = \sum_i \bar{z}_j^i \lambda_i^r \phi_i^T \approx z_{j+r}^T$.

Our approach to multifactor disentanglement is based on the following key observation: *eigenvectors of the matrix $C$ whose eigenvalue is $1$ represent time-invariant factors*. For instance, assume $C$ has a single eigenvector $\phi_1$ with $\lambda_1 = 1$ and $\lambda_i \neq 1$ for $i \neq 1$, then it follows from Eq. 4 that

$$z_j^T C^r = \bar{z}_j^1 \phi_1^T + \sum_{i=2}^k \bar{z}_j^i \lambda_i^r \phi_i^T \,. \tag{5}$$

Essentially, the contribution of $\phi_1$ is not affected by the dynamics and it remains constant, and thus the first addend remains constant throughout time, and it is related to static features of the dynamics.

In contrast, every element in the sum in Eq. 5 is scaled by its respective $\lambda_i^r$, and thus the sum changes throughout time, and these eigenvectors are related to dynamic features. We conclude that the KAE architecture virtually allows disentanglement via eigendecomposition of the Koopman matrix where the static factors are eigenvectors with eigenvalue 1, and the rest are dynamic factors.

**Multifactor Koopman Disentanglement.** Unfortunately, the vanilla KAE model is not suitable for disentanglement as the learned Koopman matrices can generally have arbitrary spectra, with multiple static factors or no static components at all. Moreover, KAE does not allow to explicitly balance the number of static vs. dynamic factors. To alleviate the shortcomings of KAE, we propose to augment the Koopman autoencoder with a spectral loss term $\mathcal{L}_{\text{eig}}$ which explicitly manipulates the structure of the Koopman spectrum, and its separation to static and dynamic factors. Formally,

$$\mathcal{L}_{\text{stat}} = \frac{1}{k_s} \sum_i^{k_s} |\lambda_i - (1 + \imath 0)|^2 \,, \tag{6}$$

$$\mathcal{L}_{\text{dyn}} = \frac{1}{k_d} \sum_i^{k_d} \xi(|\lambda_i|, \epsilon) \,, \tag{7}$$

$$\mathcal{L}_{\text{eig}} = \mathcal{L}_{\text{stat}} + \mathcal{L}_{\text{dyn}} \,, \tag{8}$$

where $k_s$ and $k_d$ represent the number of static and dynamic components, respectively, and thus $k = k_s + k_d$. The term $\mathcal{L}_{\text{stat}}$ measures the average distance of every static eigenvalue from the complex value 1. The role of $\mathcal{L}_{\text{dyn}}$ is to encourage separation between the static and dynamic factors. In practice, this is achieved with a threshold function $\xi$ which takes the modulus of $\lambda_i$ and a user parameter $\epsilon \in (0, 1)$, and it returns $|\lambda_i|$ if $|\lambda_i| > \epsilon$, and zero otherwise. Thus, $\mathcal{L}_{\text{dyn}}$ penalizes dynamic factors whose modulus is outside an $\epsilon$-ball. The inset figure shows an example spectrum we obtain using our loss penalties, where blue and red denote static and dynamic factors, respectively.

**Method Summary.** Given a batch $X \in \mathbb{R}^{b \times t \times m}$, we feed it to the encoder. Our encoder is similar to the one used in C-DSVAE (Bai et al., 2021) having five convolutional layers, followed by a uni-directional LSTM module. The output of the encoder is denoted by $Z \in \mathbb{R}^{b \times t \times k}$, and it is passed to the Koopman module. Then, $Z$ is split to past $Z_p$ and future $Z_f$ observations, allowing to compute the approximate Koopman operator via $C = Z_p^+ Z_f$. In addition, we compute $\tilde{Z}_f := Z_p \cdot C$ which will be used to compute $\mathcal{L}_{\text{pred}}$. After the Koopman module, we apply the decoder whose structure mimics the encoder but in reverse having an LSTM component and de-convolutional layers. Additional details on the encoder and decoder are detailed in Tab. 5. We decode $Z$ to obtain the reconstructed signal $X_{\text{rec}}$, and we decode $\tilde{Z}_f$ to approximate the future recovered signals $\tilde{X}_f$. The total loss is given by $\mathcal{L} = \lambda_{\text{rec}} \mathcal{L}_{\text{rec}} + \lambda_{\text{pred}} \mathcal{L}_{\text{pred}} + \lambda_{\text{eig}} \mathcal{L}_{\text{eig}}$, where the balance weights $\lambda_{\text{rec}}, \lambda_{\text{pred}}$ and $\lambda_{\text{eig}}$ scale the loss penalty terms and the exact values are given in Tab. 6. To compute $\mathcal{L}_{\text{eig}}$, we identify the static and dynamic subspaces. This is done by simply sorting the eigenvalues based on their modulus, and taking the last $k_s$ eigenvectors, whereas the rest $k_d$ are dynamic factors. Identifying multiple factors is more involved and can be obtained by manual inspection or via an automatic procedure using a pre-trained classifier, see App. B.5.

**Multifactor Static and Dynamic Swap.** Similar to previous methods our approach allows to swap between e.g., the static factors of two different input samples. In addition, our framework naturally supports multifactor swap as we describe next. For simplicity, we first consider the swap of a single factor (e.g., hair color in Sprites (Reed et al., 2015)) for the given latent codes of two samples, $z_j(u)$ and $z_j(v), j = 1, \ldots, t+1$. Denote by $\phi_1$ the eigenvector of the factor we wish to swap, then a single swap is obtained by switching the Koopman projection coefficients of $\phi_1$, i.e.,

$$\hat{z}_j(u) = \bar{z}_j^1(v)\phi_1 + \sum_{i=2}^k \bar{z}_j^i(u)\phi_i \,, \quad \hat{z}_j(v) = \bar{z}_j^1(u)\phi_1 + \sum_{i=2}^k \bar{z}_j^i(v)\phi_i \,, \tag{9}$$

where $\hat{z}_j(u)$ denotes the new code of $z_j(u)$ using the swapped factor from the $v$ sample, and similarly for $\hat{z}_j(v)$. If several factors are to be swapped, then $\hat{z}_j(u) = \sum_{i \in I} \bar{z}_j^i(v)\phi_i + \sum_{i \in I^c} \bar{z}_j^i(u)\phi_i$, where $I$ denotes the set of eigenvector indices we swap, and $I^c$ is the complement set. The above formulation

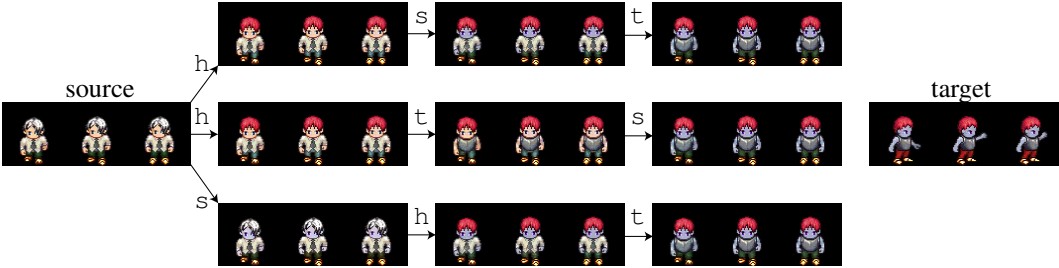

Figure 2: In the factorial swap experiment we modify individual static factors of the source character to match those of the target. The top row shows the gradual change of the hair, skin, and top colors.

is equivalent to the simpler tensor notation $\bar{Z}^s[u, :, I^c] = \bar{Z}[u, :, I^c]$ and $\bar{Z}^s[u, :, I] = \bar{Z}[v, :, I]$, where $\bar{Z} \in \mathbb{C}^{n \times t \times k}$ is the Koopman projection coefficients of a batch with $n$ samples, and $\bar{Z}^s$ represents the swapped coefficients. Thus, the swapped latent code is given by $\hat{Z} = \bar{Z}^s \cdot \Phi$, where $\Phi = (\phi_i)$ is the matrix of eigenvectors organized in columns. A more detailed description of the swaps and how to obtain the disentangled subspaces representations is provided in App. B.

## 5   RESULTS

We evaluate our model on several two- and multi-factor disentanglement tasks. For every dataset, we train our model, and for evaluation, we additionally train a vanilla classifier on the label sequences. In all experiments, we apply our model on mini-batches, extracting the latent codes $Z$ and the Koopman matrix $C$. Disentanglement tests use the eigendecomposition of $C$, where we identify the subspaces corresponding to the dynamic and static factors, denoted by $I_{\texttt{dyn}}$ and $I_{\texttt{stat}}$, respectively. We may label other subspaces such as $I_{\texttt{h}}$ to note they correspond to e.g., hair color change in Sprites. To identify the subspace corresponding to a specific factor we perform manual or automatic approaches (App. B). Importantly, subspace's dimension of a single factor may be larger than one. We provide further details regarding the network architectures, hyperparameters, datasets, data pre-processing, and a comparison of computational resources (App. B). Additional results are provided in App. C.

### 5.1   MULTIFACTOR DISENTANGLEMENT

We will demonstrate that our method disentangles sequential data to multiple distinct factors, and thus it extends the toolbox introduced in competitive sequential disentanglement approaches which only supports two factor disentanglement. Specifically, while prior techniques separate to static and dynamic factors, we show that our model identifies several semantic static factors, allowing a finer control over the factored items for downstream tasks. We perform qualitative and quantitative tasks on the Sprites (Reed et al., 2015) and MUG (Aifanti et al., 2010) datasets to show those advantages.

**Factorial swap.**   This experiment demonstrates that our method is capable of swapping individual content components between sprite characters. We extract a batch with 32 samples, and we identify by manual inspection the subspaces responsible for hair color, skin color, and top color, labeled by $I_{\texttt{h}}, I_{\texttt{s}}, I_{\texttt{t}}$. We select two samples from the test batch, shown as the source and target in Fig. 2. To swap individual static factors between the source and target, we follow Eq. 9. Specifically, we gradually change the static features of the source to be those of the target. For example, the top row in Fig. 2 shows the source being modified to have the hair color, followed by skin color, and then top color of the target, from left to right. In practice, this is achieved via setting $\bar{Z}^{\texttt{h}} = \bar{Z}^{\texttt{hs}} = \bar{Z}^{\texttt{hst}} = \bar{Z}_{\texttt{src}}$ and assigning $\bar{Z}^{\texttt{h}}[:, I_{\texttt{h}}] = \bar{Z}_{\texttt{tgt}}[:, I_{\texttt{h}}]$, $\bar{Z}^{\texttt{hs}}[:, I_{\texttt{hs}}] = \bar{Z}_{\texttt{tgt}}[:, I_{\texttt{hs}}]$, and $\bar{Z}^{\texttt{hst}}[:, I_{\texttt{hst}}] = \bar{Z}_{\texttt{tgt}}[:, I_{\texttt{hst}}]$, where $\bar{Z}_{\texttt{src}}, \bar{Z}_{\texttt{tgt}} \in \mathbb{C}^{8 \times 40}$ are the Koopman projection values of the source and target, respectively. The set $I_{\texttt{hs}} := I_{\texttt{h}} \cup I_{\texttt{s}}$, and similarly for $I_{\texttt{hst}}$. The tensor $\bar{Z}^{\texttt{h}}$ represents the new character obtained by borrowing the hair color of the target, and similarly for $\bar{Z}^{\texttt{hs}}$ and $\bar{Z}^{\texttt{hst}}$. In total, we demonstrate in Fig. 2 the changes: h→s→t (top), h→t→s (middle), and s→h→t (bottom). We additionally show in Fig. 12 an example of individual swaps including all possible combinations. Our results display good multifactor separation and transfer of individual static factors between different characters.

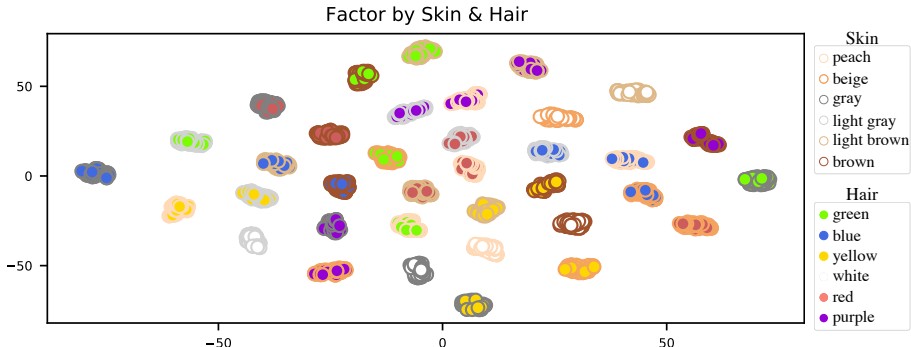

Figure 3: We show the $\mathtt{t}\text{-}\mathtt{SNE}$ plot of the 4D Koopman static subspace which encodes the skin and hair colors. The embedding perfectly clusters all (skin, hair) color combinations.

To quantitatively assess the performance of our approach in the factorial swap task, we consider the following experiment. We iterate over test batches of size 256, and for every batch we *automatically* identify its hair color and skin color subspaces, $I_{\mathrm{h}}, I_{\mathrm{s}}$. Then, we compute a random sampling of $Z$ denoted by $J$, and separately swap the hair color and the skin color. In practice, this boils down to $\bar{Z}^{\mathrm{h}} = \bar{Z}^{\mathrm{s}} = \bar{Z}$ and setting $\bar{Z}^{\mathrm{h}}[:, :, I_{\mathrm{h}}] = \bar{Z}[J, :, I_{\mathrm{h}}]$ and similarly, $\bar{Z}^{\mathrm{s}}[:, :, I_{\mathrm{s}}] = \bar{Z}[J, :, I_{\mathrm{s}}]$. The new latent codes are reconstructed and fed to the pre-trained classifier, and we compare the predicted labels to the true labels of $Z[J]$. The results are reported in Tab. 1 where we list the accuracy measures for every factor. For most non-swapped factors, we obtain an accuracy score close to random guess, e.g., the skin accuracy in the hair swap is $16.25\%$ which is very close to $1/6$. Moreover, the swapped factors yield high accuracy scores marked in bold, validating the successful swap of individual factors.

Table 1: Accuracy measures of factorial swap experiments.

| Test | action | skin | top | pants | hair |
|---|---|---|---|---|---|
| hair swap | 10.51% | 16.25% | 16.33% | 35.51% | **90.59%** |
| skin swap | 10.55% | **73.01%** | 16.29% | 30.55% | 17.70% |

**Latent Embedding.** We now explore the effect of our model on the latent representation of samples. To this end, we consider a batch $X$ of sprites where the motion, skin and hair colors are arbitrary, and the top and pants colors are fixed, for a total of 324 examples. Following the above experiment, we automatically identify the subspaces responsible for changing the hair and skin color, $I_{\mathrm{h}}, I_{\mathrm{s}}$. To explore the distribution of the latent code, we visualize the Koopman projection coefficients of the 4-dimensional subspace $I_{\mathrm{hs}} = I_{\mathrm{h}} \cup I_{\mathrm{s}}$ given by $\bar{Z}[:, :, I_{\mathrm{hs}}] \in \mathbb{C}^{324 \times 8 \times 4}$. We plot in Fig. 3 the 2D embedding obtained using $\mathtt{t}\text{-}\mathtt{SNE}$ (Van der Maaten & Hinton, 2008). To distinguish between skin and hair labels, we paint the face of every 2D point based on its true hair label, and we paint the point's edge with the true skin color. The plot resembles a grid-like pattern, showing a *perfect* separation to all 36 unique combinations of (skin, hair) colors. We conclude that the Koopman subspace $I_{\mathrm{hs}}$ indeed disentangles the samples based on either their skin or hair.

**Incremental Swap.** In this test we explore multifactor features of time-varying Koopman subspaces on the MUG dataset. Given a source image $u$, we gradually modify its dynamic factors to be those of the target $v$. In practice, we compute $\bar{Z}[u, :, I_{\mathrm{q}}] = \bar{Z}[v, :, I_{\mathrm{q}}]$, where $I_q \subset I_{\mathrm{dyn}}$ is an index set from $I_{\mathrm{dyn}}$ such that $q \in \{1, 2, 3\}$ and $I_1 \subset I_2 \subset I_3 \subset I_{\mathrm{dyn}}$. Specifically, $|I_1| = 4, |I_2| = 6, |I_3| = 32$. Fig. 4 shows the incremental swap results of two examples changing from disgust to happiness (left), and happiness to anger (right). The three rows below the source row are the reconstructions of the gradual swap denoted by $\tilde{X}(I_q) := \chi_{\mathrm{dec}}(\bar{Z}[u, :, I_q] \cdot \Phi)$. Our results demonstrate in both cases a non-trivial gradual change from the source expression to the target, as more dynamic features are swapped. For instance, the left source is mapped to a smiling character over all time samples in $\tilde{X}(I_2)$, and then it is fixed to better match the happiness trajectory source in $\tilde{X}(I_3)$.

disgust to happiness       happiness to anger

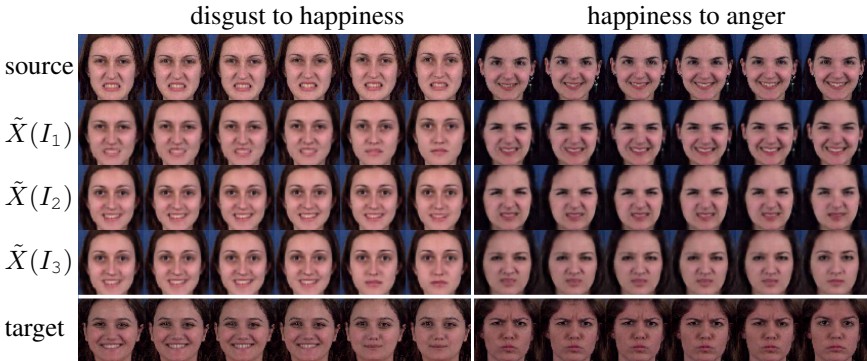

Figure 4: Our method allows to swap the dynamic features incrementally, and thus it achieves a relatively smooth transition between the source and target expressions.

## 5.2 Two Factor Disentanglement of Image Data

We perform two factor disentanglement on Sprites and MUG datasets, and we compare with state-of-the-art methods. Evaluation is performed by fixing the time-varying features of a test batch while randomly sampling its time-invariant features. Then, a pre-trained classifier generates predicted labels for the new samples while comparing them to the true labels. We use metrics such as accuracy (Acc), inception score (IS), intra-entropy $H(y|x)$ and inter-entropy $H(y)$ (Bai et al., 2021). We extract batches of size 256, and we identify their static and dynamic subspaces automatically. In contrast to most existing work, our approach is not based on a variational autoencoder model, and thus the sampling process in our approach is performed differently. Specifically, for every test sequence, we randomly sample static features by generating a new latent code based on a random sampling in the convex hull of the batch. That is, we generate random coefficients $\{\alpha_i\}$ for every sample in the batch such that they form a partition of unity and $\alpha_i \in [0, 1]$. Then, we swap the static features of the batch with those of the new samples, $\bar{Z}[:, :, I_{\texttt{stat}}] = \sum_i \alpha_i \bar{Z}[i, :, I_{\texttt{stat}}]$. We perform 300 epochs of random sampling, and we report the average results in Tab. 2, 3. Notably, our method outperforms previous SOTA methods on the Sprites dataset across all metrics. On the MUG dataset, we achieve competitive accuracy results and better results on IS and $H(y|x)$ metrics. In comparison to unsupervised methods MoCoGAN, DSVAE and R-WAE, our results are the best on all metrics.

## 5.3 Two Factor Disentanglement of Audio Data

We additionally evaluate our model on a different data modality, utilizing a benchmark downstream speaker verification task (Hsu et al., 2017) on the TIMIT dataset (Garofolo et al., 1992). In this task, we aim to distinguish between speakers, independently of the text they read. We compute for each test sample its latent representation $Z$, and its dynamic and static sub-representations $Z_{\texttt{dyn}}, Z_{\texttt{stat}}$, respectively. In an ideal two factor disentanglement, we expect $Z_{\texttt{stat}}$ to encode the speaker identity, whereas $Z_{\texttt{dyn}}$ should be agnostic to this data. To quantify the disentanglement we employ the Equal Error Rate (EER) test. Namely, we compute the cosine similarity between all pairs of latent sub-representations in $Z_{\texttt{stat}}$. The pair is assumed to encode the same speaker if their cosine similarity is higher than a threshold $\epsilon \in [0, 1]$, and the pair has different speakers otherwise. The threshold $\epsilon$ needs to be calibrated to receive the EER (Chenafa et al., 2008). If $Z_{\texttt{stat}}$ indeed holds the speaker identity,

<table>
<tr><td colspan="5">Table 2: Disentanglement metrics on Sprites.</td></tr>
<tr><td>Method</td><td>Acc↑</td><td>IS↑</td><td>$H(y|x)$↓</td><td>$H(y)$↑</td></tr>
<tr><td>MoCoGAN</td><td>92.89%</td><td>8.461</td><td>0.090</td><td>2.192</td></tr>
<tr><td>DSVAE</td><td>90.73%</td><td>8.384</td><td>0.072</td><td>2.192</td></tr>
<tr><td>R-WAE</td><td>98.98%</td><td>8.516</td><td>0.055</td><td>2.197</td></tr>
<tr><td>S3VAE</td><td>99.49%</td><td>8.637</td><td>0.041</td><td>2.197</td></tr>
<tr><td>C-DSVAE</td><td>99.99%</td><td>8.871</td><td>0.014</td><td>2.197</td></tr>
<tr><td>Ours</td><td>**100%**</td><td>**8.999**</td><td>**1.6e−7**</td><td>**2.197**</td></tr>
</table>

<table>
<tr><td colspan="5">Table 3: Disentanglement metrics on MUG.</td></tr>
<tr><td>Method</td><td>Acc↑</td><td>IS↑</td><td>$H(y|x)$↓</td><td>$H(y)$↑</td></tr>
<tr><td>MoCoGAN</td><td>63.12%</td><td>4.332</td><td>0.183</td><td>1.721</td></tr>
<tr><td>DSVAE</td><td>54.29%</td><td>3.608</td><td>0.374</td><td>1.657</td></tr>
<tr><td>R-WAE</td><td>71.25%</td><td>5.149</td><td>0.131</td><td>1.771</td></tr>
<tr><td>S3VAE</td><td>70.51%</td><td>5.136</td><td>0.135</td><td>1.760</td></tr>
<tr><td>C-DSVAE</td><td>**81.16%**</td><td>5.341</td><td>0.092</td><td>**1.775**</td></tr>
<tr><td>Ours</td><td>77.45%</td><td>**5.569**</td><td>**0.052**</td><td>1.769</td></tr>
</table>

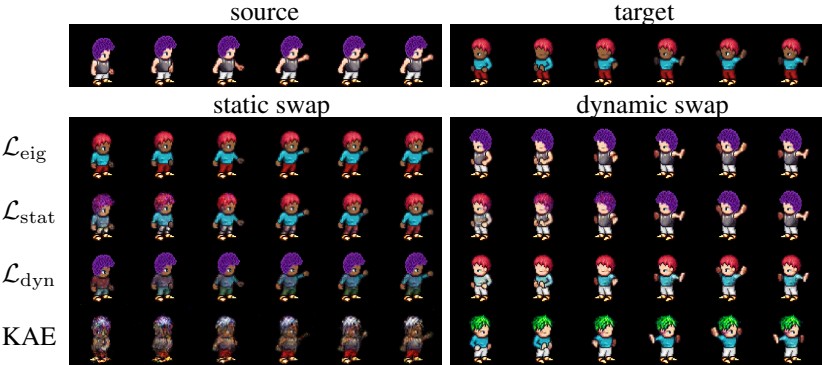

Figure 5: Our ablation study shows that the full model $\mathcal{L}_{\text{eig}}$ disentangles data well, whereas models using only $\mathcal{L}_{\text{stat}}$ loss or only $\mathcal{L}_{\text{dyn}}$ loss or no $\mathcal{L}_{\text{eig}}$ loss at all struggle with swapping static features.

then its EER score should be low. The same test is also repeated on $Z_{\text{dyn}}$ for which we expect high EER scores as it should not contain speaker information. We report the results in Tab. 4. Our method achieves the third best overall EER on the static and dynamic tests. However, S3VAE and C-DSVAE either use significantly more data or self-supervision signals. We label by C-DSVAE* and C-DSVAE† the approach C-DSVAE without content and dynamic augmentation, respectively. When comparing to unsupervised approaches that do not use additional data (FHVAE, DSVAE, and R-WAE), we achieve the best results with a margin of $0.27\%$ and $3.37\%$ static and dynamic, respectively.

Table 4: Disentanglement metrics on TIMIT.

| Method | FHVAE | DSVAE | R-WAE | S3VAE | C-DSVAE* | C-DSVAE† | C-DSVAE | Ours |
|---|---|---|---|---|---|---|---|---|
| Static EER↓ | $5.06\%$ | $5.65\%$ | $4.73\%$ | $5.02\%$ | $5.09\%$ | $4.31\%$ | $\mathbf{4.03}\%$ | $4.46\%$ |
| Dynamic EER↑ | $22.77\%$ | $19.20\%$ | $23.41\%$ | $25.51\%$ | $24.30\%$ | $31.09\%$ | $\mathbf{31.81}\%$ | $26.78\%$ |

## 5.4 ABLATION STUDY

We train different models to evaluate the effect of our loss term on the KAE architecture: full model with $\mathcal{L}_{\text{eig}}$, KAE + $\mathcal{L}_{\text{stat}}$, KAE + $\mathcal{L}_{\text{dyn}}$, and baseline KAE without $\mathcal{L}_{\text{eig}}$. All other parameters are left fixed. In Fig. 5, we show a qualitative example of static and dynamic swaps between the source and the target. Each of the bottom four rows in the plot is associated with a different model. The full model ($\mathcal{L}_{\text{eig}}$) yields clean disentanglement results on both swaps. In contrast, the static features are not perfectly swapped when removing the dynamic penalty ($\mathcal{L}_{\text{stat}}$). Moreover, the model without static loss ($\mathcal{L}_{\text{dyn}}$) does not swap the static features at all. Finally, the baseline KAE model generates somewhat random samples. We note that in all cases (even for the KAE model), the motion is swapped relatively well which can be attributed to the good encoding of the dynamics via the Koopman matrix.

## 6 DISCUSSION

We have proposed a novel approach for multifactor disentanglement of sequential data, extending existing two factor methods. Our model is based on a strong inductive bias where we assumed that the underlying dynamics can be encoded linearly. The latter assumption calls for exploiting recent Koopman autoencoders which we further enhance with a novel spectral loss term, leading to an effective disentangling model. Throughout an extensive evaluation, we have shown new disentanglement sequential tasks such as factorial swap and incremental swap. In addition, our approach achieves state-of-the-art results on two factor tasks in comparison to baseline unsupervised approaches, and it performs similarly to self-supervised and weakly-supervised techniques.

There are multiple directions for future research. First, our approach is complementary to most existing VAE approaches, and thus merging features of our method with variational sampling, mutual information and contrastive losses could be fruitful. Second, theoretical aspects such as disentanglement guarantees could be potentially shown in our framework using Koopman theory.

ACKNOWLEDGEMENTS

This research was partially supported by the Lynn and William Frankel Center of the Computer Science Department, Ben-Gurion University of the Negev, an ISF grant 668/21, an ISF equipment grant, and by the Israeli Council for Higher Education (CHE) via the Data Science Research Center, Ben-Gurion University of the Negev, Israel.

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

## A   KOOPMAN THEORY

We briefly introduce the key ingredients of Koopman theory (Koopman, 1931) which are related to our work. Consider a dynamical system $\varphi : \mathcal{M} \to \mathcal{M}$ over the domain $\mathcal{M}$ given via the update rule

$$x_{t+1} = \varphi(x_t) \,,$$

where $x_t \in \mathcal{M} \subset \mathbb{R}^m$, and $t \in \mathbb{N}$ is the time index. Koopman theory proposes an alternative representation of the dynamical system $\varphi$ by a *linear* yet infinite-dimensional Koopman operator $\mathcal{K}_\varphi$. Formally,

$$\mathcal{K}_\varphi f(x_t) = f \circ \varphi(x_t) \,,$$

where $f : \mathcal{M} \to \mathbb{C}$ is an observable complex-valued function, and $f \circ \varphi$ denotes composition of transformations. Due to the linearity of $\mathcal{K}_\varphi$, we can discuss its eigendecomposition, when it exists. Specifically, let $\lambda_j \in \mathbb{C}, \phi_j : \mathcal{M} \to \mathbb{C}$ be a pair of eigenvalue and eigenfunction respectively of $\mathcal{K}_\varphi$, i.e., it holds that

$$\mathcal{K}_\varphi \phi_j = \lambda_j \phi_j \quad \text{for any } j \,.$$

From a theoretical viewpoint, there is no loss of information to represent the dynamics with $\varphi$ or with $\mathcal{K}_\varphi$ (Eisner et al., 2015). Namely, one can recover the dynamics $\varphi$ from a given $\mathcal{K}_\varphi$ operator. Moreover, the Hartman-Grobman Theorem states that the linearization around hyperbolic fixed points is conjugate to the full, nonlinear system (Wiggins et al., 2003). The latter result was further extended to the entirety of the basin (Lan & Mezić, 2013). In practice, various tools were recently developed to approximate the infinite-dimensional Koopman operator using a finite-dimensional Koopman matrix. In particular, the Dynamic Mode Decomposition (DMD) (Schmid, 2010) is a popular technique for approximating dynamical systems and their modes. DMD was shown to be intimately related to Koopman mode decomposition in (Rowley et al., 2009), which deals with the extraction of Koopman eigenvalues and eigenfunctions in a data-driven setting. Thus, the above discussion establishes the link between our work and Koopman theory since in practice, our Koopman module is similar in spirit to DMD. Moreover, it justifies our use of the Koopman matrix to encode the dynamics as well as disentangle it.

## B   EXPERIMENTAL SETUP: ARCHITECTURE, DATASETS, HYPERPARAMETERS, AND MORE

### B.1   DATASETS

**Sprites.**   Reed et al. (2015) introduced a dataset of animated cartoon characters. Each character is composed of static and dynamic attributes. The static attributes include the color of skin, tops, pants and hair; each contains six possible variants. The dynamic attributes include three different motions: walking, casting spells and slashing, where each motion admits three different orientations: left, right, and forward. In total there are nine motions a character can perform and 1296 unique characters. A sequence is composed of eight RGB image frames of size of $64 \times 64$. We use 9000 samples for training and 2664 samples for testing.

**MUG.**   Aifanti et al. (2010) share a facial expression dataset which contains image sequences of 52 subjects. Each subject performs six facial expressions: anger, fear, disgust, happiness, sadness and surprise. Each video in the dataset consists of 50 to 160 frames. To create sequences of length 15 as described in previous work  (Bai et al., 2021), we randomly sample 15 frames from the original sequence. Then, we crop the faces using Haar Cascades face detection, and we resize to $64 \times 64$ resulting in sequences $x \in \mathbb{R}^{15 \times 3 \times 64 \times 64}$ for a total of 3429 samples. Finally, we split the dataset such that $75\%$ of it is used for the train set, and $25\%$ for the test set.

**TIMIT.**   Garofolo et al. (1992) made TIMIT available which contains 16kHz audio recordings of American English speakers reading short texts. In total, the dataset has 6300 utterances (5.4 hours) aggregated from 630 speakers reading 10 phonetically rich sentences each. For each batch of samples the data pre-processing procedure goes as follows: First, we take the maximum raw audio length in the batch, and we zero pad all samples to match that length. Second, we calculate for each sample its log spectrogram with 201 frequency features calculated by a window of 10ms, using Short Time Fourier Transform (STFT). Thus, each batch has its own $t$ (time steps) length, with an average length after padding of $t = 450$. The resulting sequences are of dimension $x \in \mathbb{R}^{t \times 201}$.

## B.2 DISENTANGLEMENT METRICS

**Accuracy** (Acc) measures how well a model preserves the fixed features while sampling the others. We compute it using a pre-trained classifier $\mathfrak{C}$ (also called judge) which is trained on the same train set and tested on the same test set as our model. The classifier outputs the probability measures per feature of the dataset. For instance, $\mathfrak{C}$ outputs one label for the pose and additional labels for each of the static factors (hair, skin, top and pants) for the Sprites dataset.

**Inception Score** (IS) measures the performance of a generator. The score is calculated by first applying the judge $\mathfrak{C}$ on every generated sequence $x_{1:t}$, yielding the conditional predicted label distribution $p(y|x_{1:t})$. Then, given the marginal predicted label distribution $p(y)$ we compute the Kullback—Leibler (KL) divergence $\mathrm{KL}\left(p(y|x_{1:t}) \mid\mid p(y)\right)$. The inception score is given by:

$$\mathrm{IS} = \exp(\mathbb{E}_x\left[\mathrm{KL}(p(y|x_{1:T})) \mid\mid p(y)]\right) .$$

**Intra-Entropy** $H(y|x)$ measures the conditional predicted label entropy of all generated sequences. To obtain the predicted labels we use the judge $\mathfrak{C}$, and we compute $\frac{1}{b}\sum_{i=1}^{b} H(p(y|x_{1:t}^i))$ where $b$ is the number of generated sequences. Lower intra-entropy score reflects higher confidence of the classifier $\mathfrak{C}$.

**Inter-Entropy** $H(y)$ measures the marginal predicted label entropy of all generated sequences. We can compute $H(p(y))$ using the judge's output on the predicted labels $\{y\}$. Higher inter-entropy score reflects higher diversity among the generated sequences.

**Equal Error Rate** (EER) is used in the speaker verification task on the TIMIT dataset. It is the value of false positive rate or false negative rate of a model over the speaker verification task, when the rates are equal.

## B.3 ARCHITECTURE AND HYPERPARAMETERS

Our models are implemented in the PyTorch (Paszke et al., 2019) framework. We used Adam optimizer (Kingma & Ba, 2014) and a learning rate of $0.001$ for all models, with no weight decay. Regarding hyper-parameters, in our experiments, $k$ is tuned between $40$ and $200$ and $\lambda_{\mathrm{rec}}, \lambda_{\mathrm{pred}}$ and $\lambda_{\mathrm{eig}}$ are tuned over $\{1, 3, 5, 10, 15, 20\}$. $k_s$ is tuned between $4$ and $20$, and the $\varepsilon$ threshold for the dynamic loss is tuned over $\{0.4, 0.5, 0.55, 0.6, 0.65\}$. The hyper-parameters are chosen through standard grid search.

### B.3.1 ENCODER AND DECODER

**Sprites and MUG.** Our encoder and decoder follow the same general structure as in Bai et al. (2021). First we have the same convolutional encoder as in C-DSVAE. Then we have a uni-directional LSTM. The architecture is described in detail in Tab. 5, where Conv2D and Conv2DT denote a 2D convolution layer and its transpose, and BN2D is a 2D batch normalization layer. Additionally, the hyperparameters are listed in Tab. 6, where $b$ is the batch size, $k$ is the size of Koopman matrix, $h$ is the dimension of the LSTM hidden state, and #epochs is the number of epochs we used for training. The balance weights $\lambda_{\mathrm{rec}}, \lambda_{\mathrm{pred}}$ and $\lambda_{\mathrm{eig}}$ scale the loss penalty terms of the Koopman layer, $\mathcal{L}_{\mathrm{rec}}, \mathcal{L}_{\mathrm{pred}}$ and $\mathcal{L}_{\mathrm{eig}}$, respectively. Finally, $k_s$ is the amount of static factors, and $\epsilon$ is the dynamic threshold, see Eqs. 6 and 7 in the main text.

Table 5: Architecture details.

| Encoder | Decoder |
|---|---|
| $64 \times 64 \times 3$ image | Z |
| Conv2D$(3, 32, 4, 2, 1) \rightarrow$ BN2D$(32) \rightarrow$ LeakyReLU | LSTM$(k, h)$ |
| Conv2D$(32, 64, 4, 2, 1) \rightarrow$ BN2D$(64) \rightarrow$ LeakyReLU | Conv2DT$(h, 256, 4, 1, 0) \rightarrow$ BN2D$(256) \rightarrow$ LeakyReLU |
| Conv2D$(64, 128, 4, 2, 1) \rightarrow$ BN2D$(128) \rightarrow$ LeakyReLU | Conv2DT$(256, 128, 4, 1, 0) \rightarrow$ BN2D$(128) \rightarrow$ LeakyReLU |
| Conv2D$(128, 256, 4, 2, 1) \rightarrow$ BN2D$(256) \rightarrow$ LeakyReLU | Conv2DT$(128, 64, 4, 1, 0) \rightarrow$ BN2D$(64) \rightarrow$ LeakyReLU |
| Conv2D$(256, k, 4, 2, 1) \rightarrow$ BN2D$(k) \rightarrow$ LeakyReLU | Conv2DT$(64, 32, 4, 1, 0) \rightarrow$ BN2D$(32) \rightarrow$ LeakyReLU |
| LSTM$(k, k)$ | Conv2DT$(32, 3, 4, 1, 0) \rightarrow$ Sigmoid |

Table 6: Hyperparameter details.

| Dataset | $b$ | $k$ | $h$ | #epochs | $\lambda_{\text{rec}}$ | $\lambda_{\text{pred}}$ | $\lambda_{\text{eig}}$ | $k_s$ | $\epsilon$ |
|---------|-----|-----|-----|---------|-----------|------------|-----------|-------|------------|
| Sprites | 32 | 40 | 40 | 800 | 15 | 1 | 1 | 8 | 0.5 |
| MUG | 16 | 40 | 100 | 1000 | 20 | 1 | 1 | 5 | 0.5 |
| TIMIT | 30 | 165 | - | 400 | 15 | 3 | 1 | 15 | 0 |

**TIMIT.** We design a neural network related to DSVAE architecture, but we use a uni-directional LSTM module instead of a bi-directional layer. The encoder LSTM input dimension is 201 which is the spectrogram features dimension and its output dimension is $k$. The decoder LSTM input dimension is $k$ and its output dimension is 201. The hyperparameter values are detailed in Tab. 6.

### B.3.2 KOOPMAN LAYER

The Koopman layer in our architecture is responsible for calculating the Koopman matrix $C$, and it is associated with the accompanying losses $\mathcal{L}_{\text{rec}}, \mathcal{L}_{\text{pred}}, \mathcal{L}_{\text{eig}}$. It may happen that the latent codes provided to the Koopman module are very similar, leading to numerically unstable computations. To alleviate this issue, we consider two possibilities. One, use blur filter on the image before inserting it to the encoder (used for the Sprites datasets). Two, add small random uniform noise sampled from $[0, 1]$ to the latent code $Z$, i.e., $Z + 0.005N$, where $N$ denotes the noise (used on TIMIT). Both options yield more diverse latent encodings, which in turn, stabilize the computation of $C$ and the training procedure. Finally, we note that our spectral penalty terms $\mathcal{L}_{\text{stat}}$ and $\mathcal{L}_{\text{dyn}}$ which compose $\mathcal{L}_{\text{eig}}$ are stable for a large regime of hyperparameter ranges.

### B.3.3 ADDITIONAL DYNAMIC LOSS OPTIONS

The proposed form of $\mathcal{L}_{\text{dyn}}$ in Eq. 7 constrains the dynamic factor modulus to an $\epsilon$-ball to promote separation between the static factors located on the point $1 + \imath 0$ and the dynamic factors. However, there are settings for which $\mathcal{L}_{\text{dyn}}$ may be not optimal. For instance, a dataset may contain measure-preserving dynamic factors, e.g., as in the motion of a pendulum. Another example includes growing dynamic factors, e.g., as in a ball moving from the center of the frame towards the boundaries of the frame. If one has additional knowledge regarding the underlying dynamics, one may adapt $\mathcal{L}_{\text{dyn}}$ accordingly. We consider the following options:

1. Set $\epsilon = 1$ while adding the dynamic loss term to $\mathcal{L}_{\text{eig}}$. In this case, $\mathcal{L}_{\text{dyn}}$ penalizes dynamic factors that are inside a $\delta$-ball around the point $1 + 0\imath$. This option addresses measure-preserving dynamic oscillations in the data.

2. Set $\epsilon = 1 + \eta, \ \eta > 0$ while adding the dynamic loss term to $\mathcal{L}_{\text{eig}}$. In this case, $\mathcal{L}_{\text{dyn}}$ penalizes dynamic factors that are inside a $\delta$-ball around the point $1 + 0\imath$. This option addresses growing dynamic factors.

### B.4 DISENTANGLEMENT PROCESS USING MULTIFACTOR DISENTANGLING KOOPMAN AUTOENCODERS

In what follows, we detail the process of extracting the multifactor latent representation of a sample, and in addition, we will demonstrate a general swap of a factor between two arbitrary samples. We let $X \in \mathbb{R}^{b \times t \times m}$ be our input batch and $x \in \mathbb{R}^m$ be a single sample that we want to calculate its multifactor disentangled latent representation. The disentanglement process of $x$ into its multiple latent factors representations using our model contains the following steps:

1. We insert $X$ into the model encoder and get the encoder output $Z \in \mathbb{R}^{b \times t \times k}$.

2. We compute the Koopman matrix $C$ for the batch $X$ using the Koopman layer as described in the main text.

3. We compute the eigendecomposition of $C$ to get the eigenvectors matrix $V \in \mathbb{C}^{k \times k}$. In addition, we calculate $U = V^{-1}$. Now we calculate $\bar{z}^T := z^T V$ for every $z \in \mathbb{R}^k$. $\bar{z}$ stores the coefficients in the Koopman space and they are the disentangled latent representation in our method. Notice that $z^T = z^T V U = \bar{z}^T U$

4. We identify the indices that correspond to each latent factor. It may be that several indices represent one factor. We use the identification method of subspaces described in B.5 to extract the indices set. Let $I_k$ be some latent factor index set. Then, the latent representation of factor $I_k$ for the input $x$ is $\bar{z}[I_k]$. For instance, $I_k$ can be the hair color factor. If we want to take a group of factors, we can aggregate a few factors together $I = I_s \cup I_t \cup I_h \cup I_p$, where $I_s = $ skin indices, $I_t = $ top indices, $I_h = $ hair indices, $I_p = $ pants indices. In practice $I$ encodes a character identity on the Sprites dataset.

To conclude, these four steps describe the process of disentangling arbitrary factors in our setup. To demonstrate a swap, let us assume we use the Sprites dataset. Let $x_1, x_2$ be two samples in $X$ and let us assume we want to swap their hair and skin attributes. We will use steps 1, 2 and 3 to extract $x_1, x_2$ multifactor latent representation $\bar{z}_1, \bar{z}_2$. Then, using Step 4, we will identify and extract $I_K = I_s \cup I_h$, were $I_h = $ hair indices and $I_s = $ skin indices. Now, we want to swap the latent representations of the hair and skin factors between the sample. To do so, we simply preform $\bar{z}_1[I_K] = \bar{z}_2[I_K]$ and vice versa $\bar{z}_2[I_K] = \bar{z}_1[I_K]$ in parallel.

To get back to the pixel space, we need to repeat our steps backward. First we need to compute the new $z_i$ after the swap. We will do it using the $V$ matrix we calculated in step 3. We compute $\tilde{z}_1^T = \bar{z}_1^T V, \tilde{z}_2^T = \bar{z}_2^T V$. Finally, we can insert $\text{Re}(\tilde{z}_1), \text{Re}(\tilde{z}_2)$ as inputs to the model decoder and get the desired swapped new samples $\tilde{x}_1, \tilde{x}_2$. Last note, if $z$ is some input for the model decoder then $z$ must be real-valued, however, $z$ is typically complex-valued since $V, U \in \mathbb{C}^{k \times k}$. Thus, we keep the real part of $z$, and we eliminate its imaginary component.

In what follows, we show that $\text{Re}(z^T V U) = z^T$, and thus feeding the real part to the decoder as mentioned above is well justified. Moreover, a similar proof holds for swapped latent vectors, i.e., $\text{Im}(\tilde{z}) = 0$. Finally, we validated that standard numerical packages such as Numpy and pyTorch satisfy this property up to machine precision.

**Theorem 1.** *If $C \in \mathbb{R}^{k \times k}$ is full rank, then $Re(z^T V U) = z^T$ for any $z \in \mathbb{R}^k$, where $V$ is the matrix of eigenvectors of $C$, and $U = V^{-1}$.*

*Proof.* It follows that

$$z^T V U = \sum_{j=1}^{k} \langle z, v_j \rangle u_j \; ,$$

where $v_j$ is the $j$-th column of $V$, and $u_j$ is the $j$-row of $U$. To prove that $\text{Im}(z^T V U) = 0$, it is sufficient to show that if $v_1$ and $v_2$ are complex conjugate pair of vectors from $V$, i.e., $v_{i_1} = \overline{v_{i_2}}$, then $\langle z, v_1 \rangle u_1$ is the complex conjugate of $\langle z, v_2 \rangle u_2$. First, we have that

$$a_1 = \langle z, v_1 \rangle = \sum_{i}^{k} z[i] v_1[i] = \sum_{i}^{k} z[i] \overline{v_2[i]} = \langle z, \overline{v_2} \rangle = \overline{a_2} \; ,$$

where the third equality holds since $v_1 = \overline{v_2}$, and the last equality holds since $z$ is real-valued. The proof is complete if we show that $u_1 = \overline{u_2}$, since then we have $\langle z, v_1 \rangle u_1 = \overline{\langle z, v_2 \rangle u_2}$. To verify that complex conjugate column pairs transform to complex conjugate row pairs, we assume w.l.o.g that the matrix $V$ can be organized such that nearby columns are complex conjugates, i.e., $v_1 = \overline{v_2}, v_3 = \overline{v_4}$, and so on. Let $P$ be the permutation matrix that exchanges the columns of $V$ to their complex conjugates, i.e., it switches the $i$-th column with the $(i+1)$-th column, where $i$ is odd. Then $VP = \overline{V}$. It follows that

$$(VP)^{-1} = P^T V^{-1} = P^T U = \overline{U} \; ,$$

namely, the $i$-th row is the complex conjugate of the $(i+1)$-th row, where $i$ is an odd number. $\qquad \square$

## B.5 Identification of Subspaces

There are two scenarios in which we need to identify semantic Koopman subspaces in the eigenvectors of the Koopman matrix $C$:

1. separate between static and dynamic information (*two factor separation*).
2. identify individual factors, e.g., hair color in sprites (*multifactor separation*).

Table 7: Accuracy measures of factorial swap experiments, see Tab. 1.

| Test | action | skin | top | pants | hair |
|------|--------|------|-----|-------|------|
| hair swap | $11.35\% \pm 0.65\%$ | $17.40\% \pm 0.79\%$ | $17.07\% \pm 0.77\%$ | $36.29\% \pm 0.88\%$ | $\mathbf{90.20\% \pm 0.52\%}$ |
| skin swap | $11.35\% \pm 0.65\%$ | $\mathbf{72.72\% \pm 0.68\%}$ | $17.23\% \pm 0.89\%$ | $31.22\% \pm 0.84\%$ | $16.92\% \pm 0.77\%$ |

**Two factor separation.** To distinguish between time-invariant and time-varying factors, we sort the eigenvalues based on their distance from the complex value $1 + \imath 0$. Then, the subspace of static features $I_{\text{stat}}$ is defined as the eigenvalues' indices of the first $k_s$ elements in the sorted array. Then, the dynamic features subspace $I_{\text{dyn}}$ holds the remaining indices, i.e., $I_{\text{dyn}} = I \setminus I_{\text{stat}}$, where $I$ is the set of all indices, and $S_1 \setminus S_2$ generates the set difference of the sets $S_1$ and $S_2$.

**Multifactor separation.** The identification of individual features such as the hair color or skin color in Sprites is less straightforward, unfortunately. Essentially, the key difficulty lies in that the Koopman matrix may encode an individual factor using a subspace whose dimension is unknown a priori. In addition, the subspace related to e.g., hair color may depend on the particular batch sample. For instance, we observed cases where the hair color subspace was of dimension $1, 2$ and $3$ for three different batches. Nevertheless, manual inspection of $I_{\text{stat}}$ typically reveals the role of the eigenfunctions, and it can be achieved efficiently as $k_s \leq 15$ in our experiments. Still, we opt for an automatic approach, and thus we propose the following simple procedure. We consider the power set of $I_{\text{stat}}$, denoted by $\mathcal{I}_{\text{stat}}$. Let $J$ be an element of $\mathcal{I}_{\text{stat}}$, then we swap the content of the batch with respect to $J$, and check the accuracy of the factor in question (e.g., hair color) using the pre-trained classifier. The subspace $J$ which corresponds to a single factor change is the one for which the accuracy of the factor decreases the most with respect to the original samples. In practice, we noticed that often the subspace of a factor is composed of subsequent eigenvectors in the sorting described for the two factor separation. Thus, many subsets $J$ of the power set $\mathcal{I}_{\text{stat}}$ can be ignored. We leave further exploration of this aspect for future work.

### B.6 SPEAKER VERIFICATION EXPERIMENT DETAILS

The speaker verification task in Sec. 5.3 is performed as follows. We use the test set of TIMIT which contains 24 unique speakers, with eight different sentences per speaker. In total there are 192 audio samples. We compute the latent representation $Z$ for this data, and its Koopman matrix $C$. Using the eigendecomposition of $C$, we identify the static and dynamic subspaces $I_{\text{stat}}$ and $I_{\text{dyn}}$. We denote by $Z_{\text{stat}}, Z_{\text{dyn}}$ the latent codes obtained when projecting $Z$ to $I_{\text{stat}}, I_{\text{dyn}}$, respectively. Formally, this is computed via $Z_{\text{stat}} = Z \cdot \Phi^{-1}[:, I_{\text{stat}}] \cdot \Phi[I_{\text{stat}}]$, and similarly for the dynamic features. To perform the speaker verification task we calculate the identity representation code for the batch given by

$$\hat{Z}_{\text{stat}} = \frac{1}{t} \sum_{j=1}^{t} Z_{\text{stat}}[:, j, :], \quad \hat{Z}_{\text{dyn}} = \frac{1}{t} \sum_{j=1}^{t} Z_{\text{stat}}[:, j, :], \quad \hat{Z}_{\text{dyn}}, \hat{Z}_{\text{dyn}} \in \mathbb{R}^{192 \times 165}.$$

The EER calculations are performed separately for $\hat{Z}_{\text{stat}}$ and $\hat{Z}_{\text{dyn}}$ for all of their $192^2 = 18336$ pair combinations.

## C ADDITIONAL RESULTS

### C.1 MEAN AND STANDARD DEVIATION MEASURES

We report the mean and standard deviation measures computed over 300 runs for the results reported in Tab. 1, 2, 3 in the main text. The results are detailed in Tab. 7, 8. The low standard deviation highlights the robustness of our method to various seed numbers, and the overall stability of our trained models.

### C.2 DATA GENERATION

We present qualitative results of our model's unconditional generation capabilities. To this end, we randomly sample static and dynamic features by producing a new latent code based on a random

Table 8: Disentanglement metrics on Sprites and MUG, see Tabs. 2 3.

| Method | Acc↑ | IS↑ | $H(y|x)\downarrow$ | $H(y)\uparrow$ |
|---|---|---|---|---|
| Sprites | $100\% \pm 0\%$ | $8.999 \pm 2.3\mathrm{e}{-6}$ | $1.6\mathrm{e}{-7} \pm 2.2\mathrm{e}{-7}$ | $2.197 \pm 0$ |
| MUG | $77.45\% \pm 0.62\%$ | $5.569 \pm 0.026$ | $0.052 \pm 0.004$ | $1.769 \pm 0$ |

sampling in the convex hull of two randomly chosen samples from the batch. That is, for every sample in the batch we generate random coefficients $\{\alpha_j \in [0,1]\}$ which form a partition of unity $\sum_{j \in J} \alpha_j = 1$, where $J$ denotes the sample indices, and $|J| = b = 2$ is the number of samples in the combination. Then, we swap the static or dynamic features of the source (src) sample using the convex combination, $\bar{Z}[\text{src}, :, I_{\text{stat}}] = \sum_{j \in J} \alpha_j \bar{Z}[j, :, I_{\text{stat}}]$, $\bar{Z}[\text{src}, :, I_{\text{dyn}}] = \sum_{j \in J} \alpha_j \bar{Z}[j, :, I_{\text{dyn}}]$, respectively. The reconstruction of the latent codes for which static or dynamic factors are swapped are shown on the right panels in Figs. 6, 7, 13, 14 respectively. Our results on both Sprites and MUG datasets demonstrate a non-trivial generation of factors while preserving the dynamic/static factors shown on the left panels.

### C.3 TWO FACTORS AND MULTIFACTOR SWAPS

We present several qualitative results of two factor swapping between static and dynamic factors of two given samples. In Figs. 8 and 9, each odd indexed row $i \in \{1, 3, 5, 7\}$ shows the source sequence on the left and the target sequence to the right. Even indexed rows $j \in \{2, 4, 6, 8\}$ represent the reconstructed samples after the swap where on the left we show the static swap, and on the right the dynamic swap. Notably, all examples show clean swaps while preserving non-swapped features.

Additionally, we extend the result in Fig. 2 to show an example in which we swap all multifactor combinations. Specifically, we show in Fig. 12 several multifactor swap from the source sequence (top row) to the target sequence (bottom row). The text to the left of every sequence in between denotes the swapped factor(s). For instance, the second row with the text p shows how the pants color of the target is swapped to the source character. Similarly, the row with the text s+t+h is related to the swap of the skin, top, and hair colors.

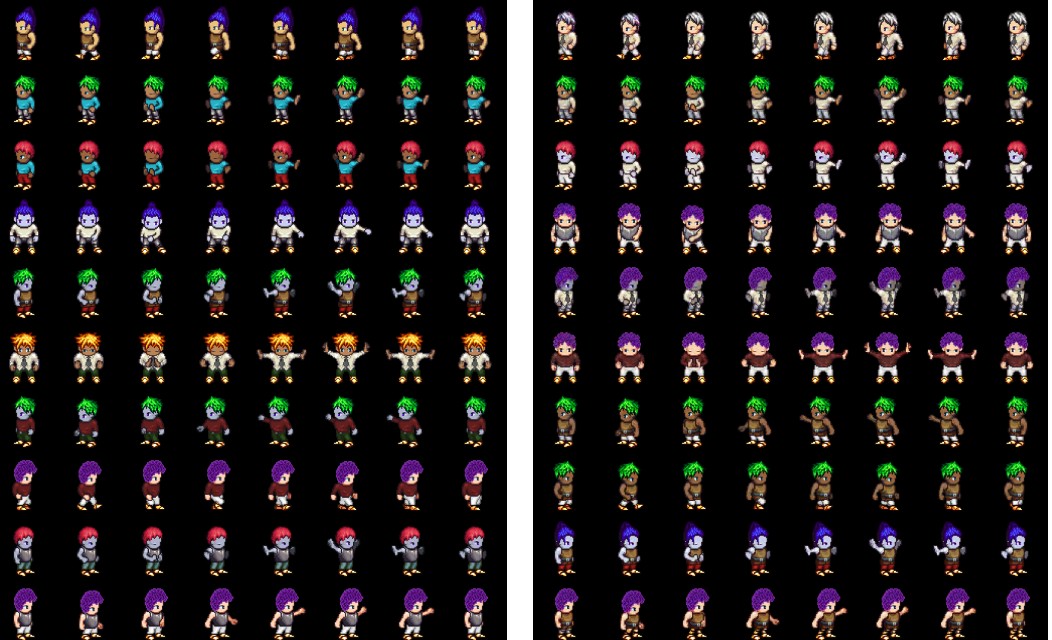

Figure 6: Unconditional generation of Sprite characters. The left panel shows the source sequences, and the right panel demonstrates the sampled characters where time-varying features are preserved.

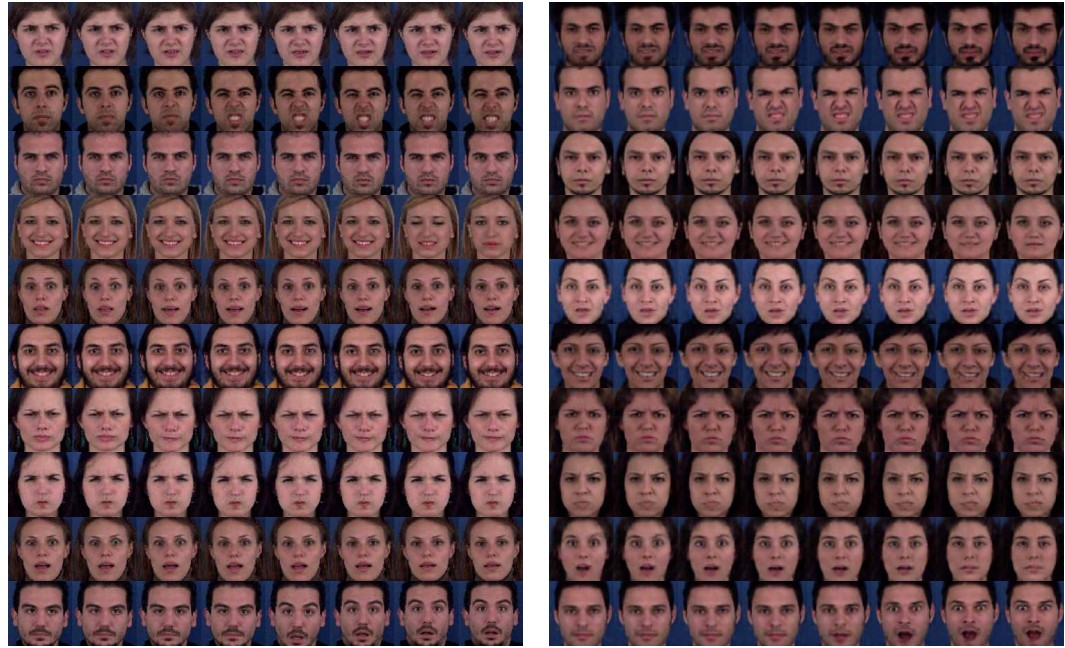

Figure 7: Unconditional generation for the MUG dataset. The left panel shows the source sequences, and the right panel demonstrates the sampled identities where time-varying features are preserved.

## C.4 STATIC INCREMENTAL SWAP ON MUG

Similarly to Fig. 4 in the main text, we now show an incremental swap example on the MUG dataset where the static features are swapped gradually, see Fig. 10. The multifactor subspaces used in this experiment are of sizes $|I_1| = 1, |I_2| = 2, |I_3| = 5$ where $I_1 \subset I_2 \subset I_3 \subset I_{\text{stat}}$. We observe a non-trivial gradual change from the source sequence (top row) to the target sequence (bottom row). In each incremental step, more static features are changing towards the target samples. Specifically, the skin color, hair color and density, ears structure, nose structure, chicks structure, chicks texture, lips and more other physical characteristics change gradually to better match the physical appearance of the target. Additionally, we observe that the source expression of the source is not altered during the transformation, highlighting the disentanglement capabilities of our approach.

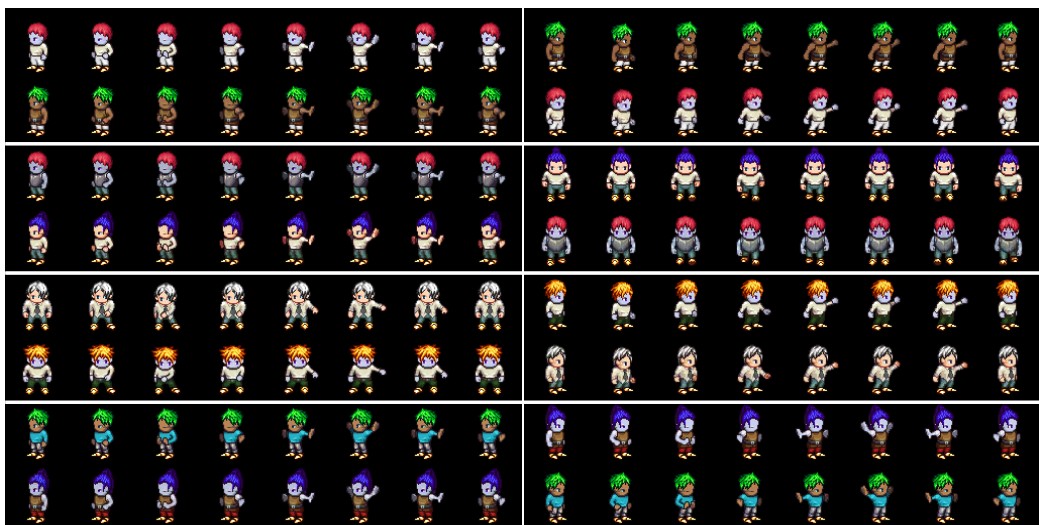

Figure 8: Several static and dynamic swap results on the Sprites dataset.

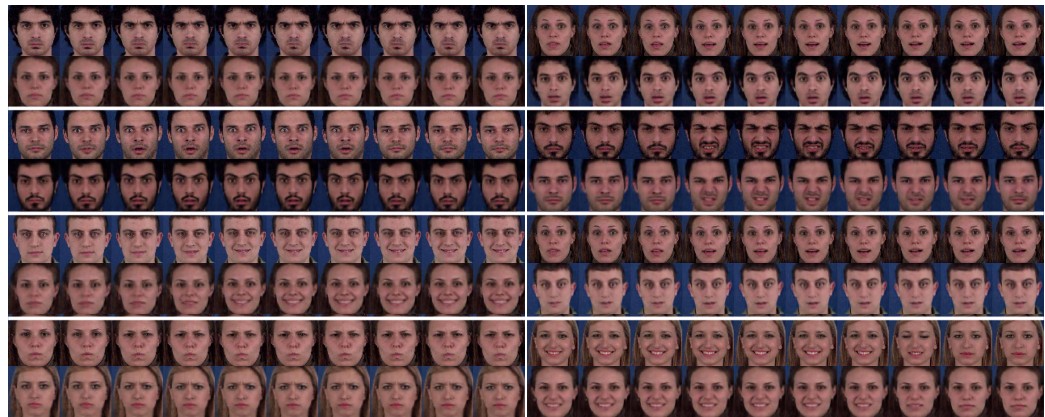

Figure 9: Several static and dynamic swap results on the MUG dataset.

## C.5  KOOPMAN MATRIX SPECTRUM ABLATION STUDY

We would like to explore the impact of our spectral loss on the spectrum and the eigenvalues scattering of the Koopman matrix $C$. To this end, we train four different models: full model with $\mathcal{L}_{\text{eig}}$, KAE + $\mathcal{L}_{\text{stat}}$, KAE + $\mathcal{L}_{\text{dyn}}$, and baseline KAE without $\mathcal{L}_{\text{eig}}$. We show in Fig. 11 the obtained spectra for the various models, where eigenvalues associated with static factors are marked in blue, and the dynamic components are highlighted in red. Our model shows a clear separation between the static and dynamic factors, allowing to easily disentangle the data in practice. In contrast, the models KAE and $\mathcal{L}_{\text{stat}}$ yield spectra in which the static and dynamic components are very close to each other, leading to challenging disentanglement. Finally, the model $\mathcal{L}_{\text{dyn}}$ shows separation in its spectrum, however, some of the static factors drift away from the eigenvalue 1.

## C.6  COMPUTATIONAL RESOURCES COMPARISON

We compare our method in terms of network memory footprint and the amount of data used for the Sprites dataset. We show in Tab. 9 the comparison of our method with respect to the other methods. All other approaches use significantly more parameters than our method, which uses 2 million weights. In addition, S3VAE and C-DSVAE utilize additional information during training. S3VAE exploits supervisory signals to an unknown extent as the details do not appear in the paper, and the code is proprietary. C-DSVAE uses data augmentation of size sixteen times the train set, that is, for content augmentation they generated eight times more train data, and the same amount for the motion augmentation. In comparison, our method and DSVAE do not use any additional data on top of the train set.

The time complexity analysis of our method is governed by the complexities of the encoder, decoder, the Koopman layer and the loss function. The encoder and decoder can be chosen freely and are

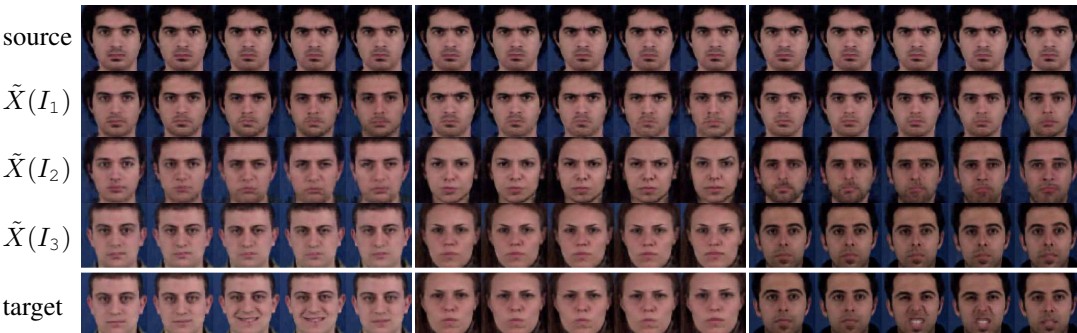

Figure 10: An incremental swap result of the static features on the MUG dataset.

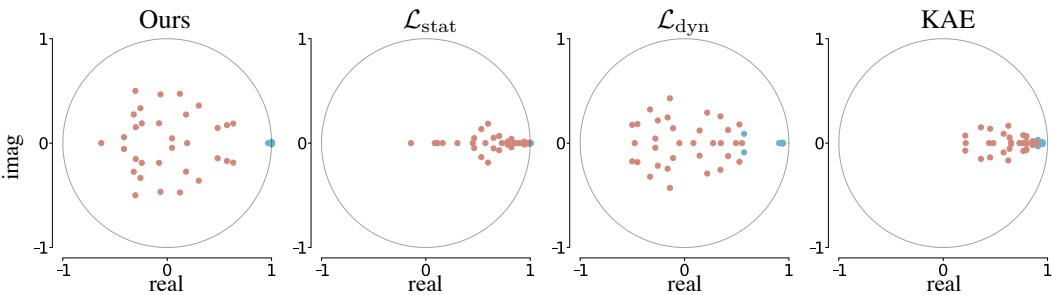

Figure 11: The Koopman matrix spectrum of different models.

typically similar to prior work (Hsu et al., 2017; Li & Mandt, 2018; Zhu et al., 2020; Bai et al., 2021), and thus we focus our analysis on the Koopman layer and the loss function. The dominant operation in the Koopman layer in terms of complexity is the computation of the pseudo-inverse of $Z_p$ (please see Section 3). Computing the pseudo-inverse of a matrix is implemented in high-level deep learning frameworks such as pyTorch via SVD. The textbook complexity of SVD is $\mathcal{O}(\min(mn^2, m^2n))$ for an $m \times n$ matrix. In addition, computing the loss function involves eigendecomposition. The theoretic complexity of eigendecomposition is equivalent to that of matrix multiplication, which in our case is $\mathcal{O}(k^{2.376})$, where the Koopman operator is of size $k \times k$. In comparison, the matrices $Z_p$ for which we compute pseudo-inverse are of size $b \cdot t \times k$, and typically $k < b \cdot t$. Thus, the pseudo-inverse operation governs the complexity of the algorithm. The development of efficient SVD algorithms for the GPU is an ongoing research topic in itself. As far as we know, there is some parallelization in torch SVD computation, mainly affecting the decomposition of *large* matrices. The Koopman matrices we use are typically small (e.g., $100 \times 100$), and thus the effective computation time is short.

Table 9: Computational resources comparison.

| Method | DSVAE | R-WAE | S3VAE | C-DSVAE | Ours |
|---|---|---|---|---|---|
| Type | unsupervised | (weakly) unsupervised | self-supervised | self-supervised | unsupervised |
| Params | 21M | 121M | 11M | 11M | 2M |
| Data | - | labels | supervisory signals | data augmentation ($\times 16$) | - |

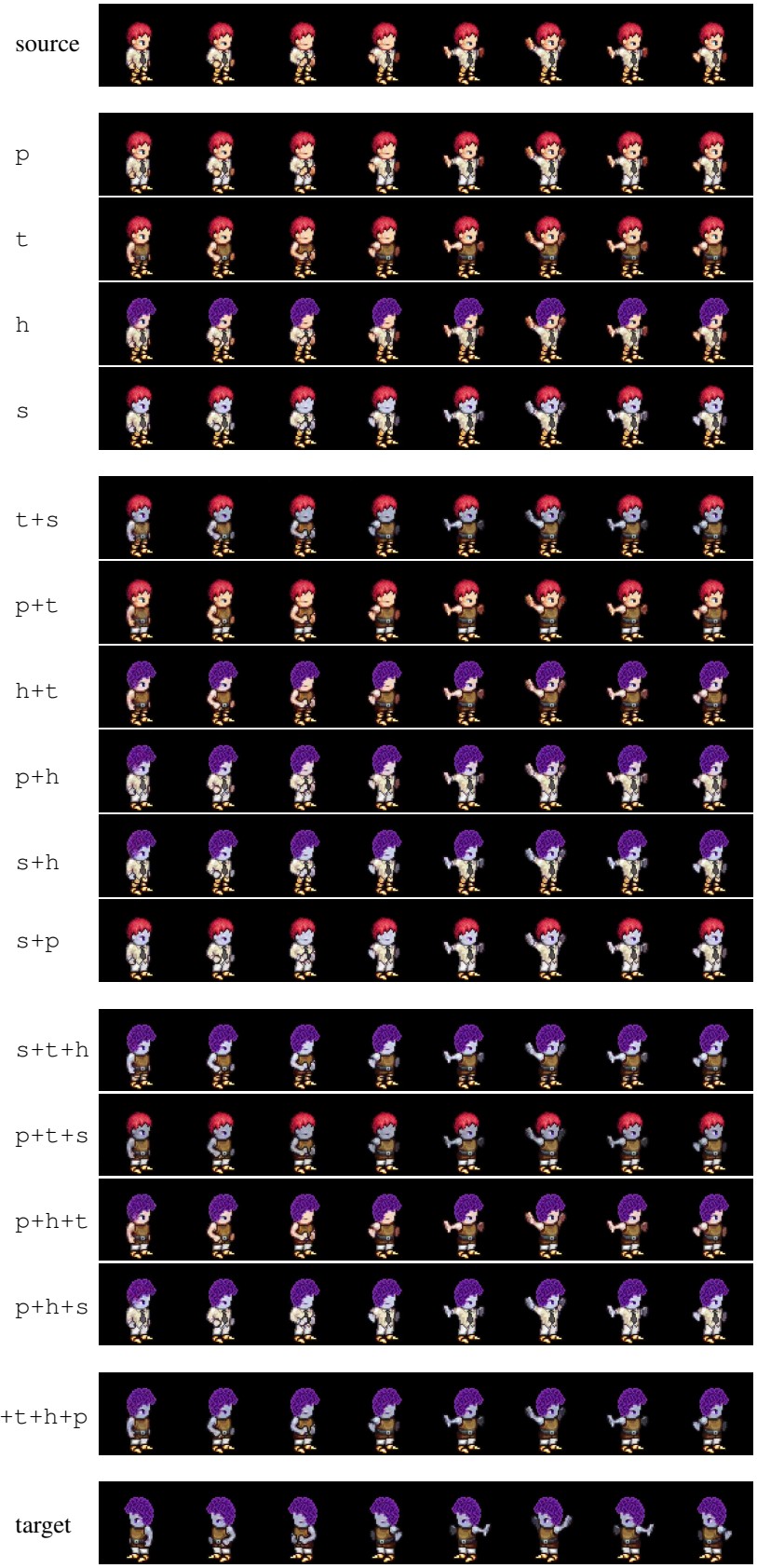

Figure 12: Multifactor swap of individual static factors and their combinations on the Sprites dataset.

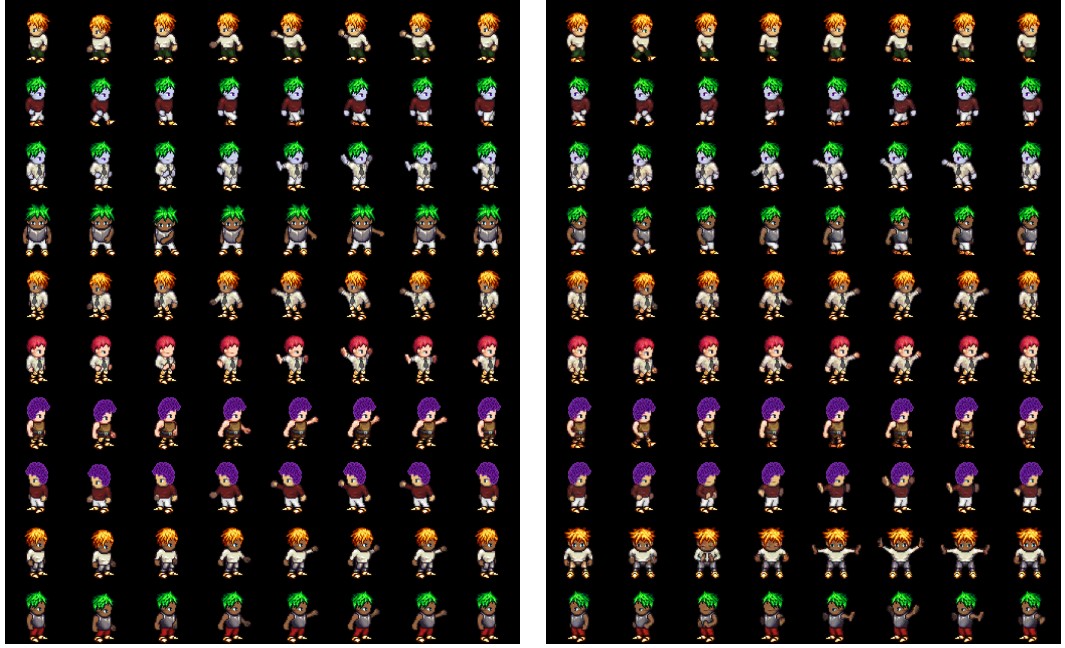

Figure 13: Unconditional generation of Sprite characters where the static factors are kept fixed, and the dynamic features are randomly sampled.

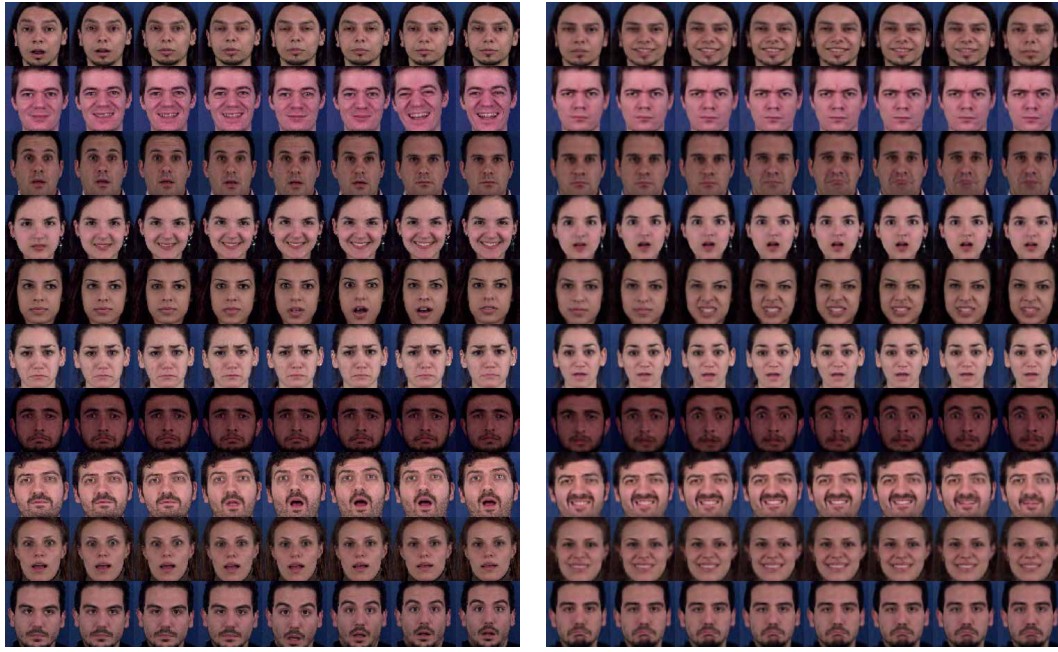

Figure 14: Unconditional generation of MUG images where the static factors are kept fixed, and the dynamic features are randomly sampled.

