# OpenReview forum: "Multifactor Sequential Disentanglement via Structured Koopman Autoencoders"
_ICLR.cc/2023/Conference — ICLR 2023 notable top 25%_

### Official Review · Reviewer_WaMN · 2022-10-23

**Confidence:** 3
**Correctness:** 4
**Technical Novelty And Significance:** 3
**Empirical Novelty And Significance:** 3
**Recommendation:** 8

**Clarity, Quality, Novelty And Reproducibility:**

To my knowledge the work is original. The writing and experimental results are clear, and the information provided in the paper and appendix is pretty comprehensive.

**Strength And Weaknesses:**

Strengths:
- To my knowledge the work is a novel contribution to two field (Koopman-based modeling and disentangling of complex data)
- Experiments were interesting, relatively extensive, and served as a good illustration of the benefits of the proposed method

Weaknesses:
- The process of multifactor separation seems potentially cumbersome and I'm not sure how easily it could be extended to different problems.
- A few questions about the algorithm itself that should be clarified.

**Summary Of The Paper:**

This paper presents a method for disentangling data into latent factors of variation by enforcing structure within Koopman matrices used to model the data's time evolution. This procedure allows for disentangling the static and dynamic components of the data, and furthermore disentangling features within those components (e.g. hair color, skin color, etc.). Experiments show the ability to use trained models to swap features within images and improve on existing approaches in benchmark tasks.

**Summary Of The Review:**

I found this paper to be interesting and of good quality. I thought this was a good extension to prior Koopman-based modeling approaches and seems to provide value within the field of data disentangling.

A few comments/questions:
- In the loss penalty applied to the dynamic eigenvalues we seem to be explicitly imposing that the eigenvalues are less than one. In a dynamical system, this corresponds to having no unstable modes. For image data, how would having an "unstable" mode with eigenvalue magnitude >1 manifest itself? Do you ever see this in cases where the loss penalty isn't applied?
- This paper is working with datasets where there are clear distinctions between attributes within the dataset (especially in the sprite dataset). Can you set the number of static modes to be the number of factors that vary within the dataset (i.e. if only hair color, skin color, and shirt color vary, can you get away with only using three static modes)? If not, why do you think that is?
- In dimensionality reduction techniques that rely upon SVD, it is usually assumed that the modes with the largest eigenvalues are most important, i.e. they explain most of the variance in the data. Did you see anything similar in your experiments? If you were working with a dataset for which you had less a priori knowledge about important attributes, do you think you would be able to discover them through looking at the dominant modes?
- What role does the LSTM play in the encoder architecture? It would be possible to perform dimensionality reduction with the convolutional layers alone. Given that the temporal relationship between encodings is being modeled by the Koopman operator and not the LSTM, why do you need to run the encodings through an LSTM first?
- In the appendix it states "Thus, we keep the real part of z, and we eliminate its imaginary component". This seems like it would be throwing out some information? Why do you think this is justified, and did you try alternatives like calculating the magnitude, feeding both the real and imaginary components to the decoder, etc.?

---

> ### Author Response · Authors · 2022-11-16
> **Response to Reviewer WaMN - Part1**
>
> We would like to thank Reviewer WaMN for positively commenting on the novelty of our work, the extensiveness of our experiments, and our clear writing. In addition, we would like to thank them for challenging some of our assumptions and raising important questions that help us improving our paper. Below, we address the comments raised by Reviewer WaMN. Given the opportunity, we will be happy to incorporate the modifications listed below into a final revision.
>
> > The process of multifactor separation seems potentially cumbersome and I'm not sure how easily it could be extended to different problems.
>
> We strive to make our approach as accessible as possible to interested practitioners and researchers. We believe that incorporating the suggestions made by the reviewers will improve and simplify the discussion in the paper regrading multifactor separation. In addition, we will upload a full git repository of Koopman Disentanglement to facilitate the reproduction of our results, and to inspire future research in the field. The package will include the models, the disentanglement process, the subspace search methods and more components that we believe will make it easier to use for multifactor disentanglement. We hope that our paper with its code repository will provide a solid framework for Koopman-based  disentanglement learning.
>
> > In the loss penalty applied to the dynamic eigenvalues we seem to be explicitly imposing that the eigenvalues are less than one. In a dynamical system, this corresponds to having no unstable modes. For image data, how would having an "unstable" mode with eigenvalue magnitude >1 manifest itself? Do you ever see this in cases where the loss penalty isn't applied?
>
> To answer the first question, yes, unstable modes can also appear in image data. Consider the opposite example to the one we suggested in our response to Reviewer tz1i. Namely, say we have a sequence of images with a ball moving in a spiral motion from the center of the image towards its boundaries. There are two dynamic factors: 1. the angle, and 2. the distance from the center. The second factor is "unstable" as it grows over time until the ball leaves the frame. Theoretically, we expect that it would be a manifestation of an eigenvalue $>1$. Regarding the second question, no, we did not observe such cases in our ablation study when the dynamic loss was not used. The modes are still "stable" (i.e., eigenvalue $\leq 1$). It is possible that unstable features are not dominant in the datasets we considered.
>
> > This paper is working with datasets where there are clear distinctions between attributes within the dataset (especially in the sprite dataset). Can you set the number of static modes to be the number of factors that vary within the dataset (i.e. if only hair color, skin color, and shirt color vary, can you get away with only using three static modes)? If not, why do you think that is?
>
> This is an interesting question, and it is somewhat related to the question raised by Reviewer tz1i regarding multifactor disentanglement. Theoretically, one should be able to fix the number of static factors $k_s$ to match the actual static variations in the data, assuming this knowledge is available. However, we observed in our experiments that fixing $k_s$ in this way yields inferior multifactor disentanglement results. We believe that there are two potential causes which may be related here: 1. Koopman eigenvectors are typically not orthogonal, and 2. the static factors are not expressive enough to capture static information. In practice, we observed "bleeding" of information between factors, and the collapse of variations into the same group of factors. In our paper we solved the above issues by increasing $k_s$. However, incorporating mutual information penalties as discussed in the response to Reviewer tz1i may also be fruitful.
>
> >  In dimensionality reduction techniques that rely upon SVD, it is usually assumed that the modes with the largest eigenvalues are most important, i.e. they explain most of the variance in the data. Did you see anything similar in your experiments? If you were working with a dataset for which you had less a priori knowledge about important attributes, do you think you would be able to discover them through looking at the dominant modes?
>
> In our framework, there is no real notion of dominant modes, as all modes are important. We separate factors to static and dynamic based on their eigenvalues properties, but, all factors are equally important and used during decomposition and transfer of features. Also, we would like to mention that the attributes are learned in an unsupervised fashion based on the penalty losses and our Koopman modeling. We do not incorporate any a priori knowledge on important attributes of the dataset when we perform training and inference with our framework. We are unsure whether our response accurately answers the question. We would be happy to extend this further if necessary.

---

> > ### Comment · Reviewer_WaMN · 2022-11-18
> > **Thank you**
> >
> > Thank you for the responses!

---

> ### Author Response · Authors · 2022-11-16
> **Response to Reviewer WaMN - Part2**
>
> > What role does the LSTM play in the encoder architecture? It would be possible to perform dimensionality reduction with the convolutional layers alone. Given that the temporal relationship between encodings is being modeled by the Koopman operator and not the LSTM, why do you need to run the encodings through an LSTM first?
>
> Koopman approaches based on our discussion in Appendix A are autonomous, i.e., they require that information needed to predict the next observable is in the current observable. In practice, an encoder that transforms states while ignoring their temporal relations will struggle to encode all necessary information such that $z_{t+1}^T \approx z_t^T C$. We use an LSTM component to better encode past information into the latent code $z_t$, so that the resulting Koopman system will be autonomous. We are happy to add this discussion to the final revision, if the reviewer thinks it is important.
>
> > In the appendix it states "Thus, we keep the real part of z, and we eliminate its imaginary component". This seems like it would be throwing out some information? Why do you think this is justified, and did you try alternatives like calculating the magnitude, feeding both the real and imaginary components to the decoder, etc.?
>
> Thank you for giving us the opportunity to elaborate on this point. In theory and practice, there is no loss of information when taking the real part of $z$. From a theoretical point of view, we can prove that our decomposition of $z$ (see Eq. 4) maintains complex conjugate pairs, and thus the imaginary part cancels. Further, we empirically checked that Numpy and pyTorch implementations respect this property (up to machine precision). We will add the proof and a short discussion to Appendix B.4.

---

### Official Review · Reviewer_oLc4 · 2022-10-23

**Confidence:** 4
**Correctness:** 4
**Technical Novelty And Significance:** 3
**Empirical Novelty And Significance:** 3
**Recommendation:** 6

**Clarity, Quality, Novelty And Reproducibility:**

This is the first work that introduces Koopman theory into sequential disentanglement problem. In general, the paper is written clearly and the experimental results seem to be promising.

**Strength And Weaknesses:**

Pros:
1. Handling multi-factors is a very tough challenge. This paper proposes a very interesting idea to address this problem.
2. The experiments are thorough and the performance looks promising.
3. The structure of the paper is clear and informative.

Cons:
1. How do you decide length T?
2. While I think the claim "eigenvectors of the matrix C whose eigenvalue is 1 represent time-invariant factors" does make sense, how do you set thresholds and decide whether the eigenvalue is close to 1 or not?
3. How costly is the whole eigendecomposition?
4. Deriving a new C for each Z might cause overfitting. Have you considered stationary assumptions?
5. For more real-world problems where there might not be explicit semantic meanings, how do you map the eigenvectors to semantics?
6. Exploring the predictiveness of past for future has been seen in some prior works like predictive information [1,2]. You can also discuss the similarities and differences.

[1] Clark, D., Livezey, J. and Bouchard, K., 2019. Unsupervised discovery of temporal structure in noisy data with dynamical components analysis. Advances in Neural Information Processing Systems, 32.

[2] Bai, J., Wang, W., Zhou, Y. and Xiong, C., 2020, September. Representation Learning for Sequence Data with Deep Autoencoding Predictive Components. In International Conference on Learning Representations.

**Summary Of The Paper:**

The paper proposes a new autoencoder method for sequence disentanglement. This work extends the prior works to handle multi-factor sequential disentanglement. While keeping the general encoder and decoder structure, the model adds a Koopman module for the latent space. This work holds the assumption that the underlying dynamics can be represented linearly in the latent space. The Koopman layer computes the Koopman matrix and uses eigenvectors as static (eigenvalue 1) or dynamic factors (others).

**Summary Of The Review:**

This is a novel work with noticeable contributions. The goal is clear and the work improves over existing works. But the computation cost and some details (hyper-params, thresholds) can be elaborated on.

---

> ### Author Response · Authors · 2022-11-16
> **Response to Reviewer oLc4**
>
> We would like to thank Reviewer oLc4 for acknowledging the toughness of problem, the interesting approach we suggest to solve it, the extensive experiments conducted to evaluate the method, and the clarity of the writing. We also would like to thank them for their detailed comments, questions and suggestions that help to deepen our discussion and improve the paper. Below, we address the comments raised by Reviewer oLc4. Given the opportunity, we will be happy to incorporate the modifications listed below into a final revision.
>
> > How do you decide length T?
>
> The length of the sequence $T$ for the different datasets is taken by the same protocol used e.g., in DSVAE (Li \& Mandt 2018) C-DSVAE (Bai et al. 2021) for a fair comparison, and to align with existing benchmarks. We added a clarification to the revision, including the particular length $T$ for each dataset.
>
> > While I think the claim "eigenvectors of the matrix C whose eigenvalue is 1 represent time-invariant factors" does make sense, how do you set thresholds and decide whether the eigenvalue is close to 1 or not?
>
> Thank you for this comment. We do not set an explicit threshold to decide whether an eigenvalue is close to 1. Instead, we use the hyperparameter $k_s$ for the number of static factors in the data, chosen through standard grid search. Thus, in order to decide which components are close to 1, we take the $k_s$ nearest components. Practically, we observe that the components we chose are close to 1 ($\approx.99$). We elaborate our discussion regarding hyperparameters in the revision.
>
> > How costly is the whole eigendecomposition?
>
> The theoretic complexity of eigendecomposition is equivalent to that of matrix multiplication, which in our case is $\mathcal{O}(k^{2.376})$, where the Koopman operator is of size $k \times k$ (see e.g., Fast Linear Algebra is Stable by Demmel et al.). In comparison, the matrices $Z_p$ for which we compute pseudo-inverse are of size $b\cdot t \times k$, and typically $k < b \cdot t$. Thus, the pseudo-inverse operation (computed via SVD, see the discussion in our appendix C.6) governs the complexity of the algorithm. We will incorporate the latter discussion to appendix C.6 in our revision.
>
> > Deriving a new C for each Z might cause overfitting. Have you considered stationary assumptions?
>
> First, we want to clarify how we avoid overfitting in our case. Indeed, deriving a new $C$ for each $Z$ might cause overfitting in the general setting. However, in our setting, $Z \in \mathbb{R}^{b \cdot t \times k}$ where $b \cdot t$ is much greater than $k$. Thus, when computing $C$, we have more constraints than variables, yielding an approximate Koopman operator fitting the data in a least squares sense. Second, regarding stationary assumptions, we would like to note that our model is *deterministic*, and thus to consider stationary assumptions we need to extend our model to the probabilistic setting. This is an interesting idea and might be a good direction for future work and research.
>
> > For more real-world problems where there might not be explicit semantic meanings, how do you map the eigenvectors to semantics?
>
> We would like to note that for the MUG and TIMIT datasets, the semantic information is less explicit in comparison to SPRITES. In general, we do not enforce any specific mapping between the semantics that might be in the dataset and the different eigenvectors. Instead, our method learns the semantics in an unsupervised fashion, based on the constraints we employ via the loss functions and Koopman modeling. Our empirical evaluation on standard sequential benchmarks shows that if semantic structures exist in the data, then our method can identify and disentangle them effectively.
>
> > Exploring the predictiveness of past for future has been seen in some prior works like predictive information [1,2]. You can also discuss the similarities and differences.
>
> We are thankful for your suggestion to deepen our discussion about other methods. We added a discussion about similarities and differences in comparison to the suggested methods in the revision.

---

### Official Review · Reviewer_tz1i · 2022-10-24

**Confidence:** 4
**Correctness:** 3
**Technical Novelty And Significance:** 4
**Empirical Novelty And Significance:** 3
**Recommendation:** 8

**Clarity, Quality, Novelty And Reproducibility:**

The writing is very good. The experiments nicely support the claim of the paper. The method is at least technically novel to some extent, while I cannot really assess the significance of the empirical results in terms of disentanglement research. The results seem to be reproducible with the attached codes, while I did not try them by myself.

**Strength And Weaknesses:**

### Strengths

- The idea of using Koopman operator's spectrum for static/dynamic disentanglement is very interesting and reasonable.
- The proposed method is examined against different datasets with different evaluation metrics.

### Weaknesses

1. Despite the nice reasoning for static/dynamic disentanglement, multifactor disentanglement is not well supported by theoretical arguments. I do not think it is a fatal flaw, but it is certainly a weakness as one of the main claims of the paper is the capability of multifactor disentanglement.

2. The definition of $\mathcal{L}_\text{dyn}$ penalize $| \lambda |$ regardless of $\angle \lambda$. This sounds strange because even if $\vert \lambda \vert=1$, dynamic modes with $\angle \lambda \neq 0$ represent (energy-preserving) oscillations, which would be regarded as dynamic factors. I guess, in the presented experiments, the current definition of $\mathcal{L}_\text{dyn}$ worked as expected only because such energy-preserving oscillations were not dominant in the datasets.

### Other comments, not necessarily weaknesses

3. Multifactor-ness only holds for static factors. While it might be a limit of the proposed method, it is understandable because, in many datasets, dynamic factors correspond to dissipative dynamic modes (with, say, $\vert \lambda \vert < 0.9$) that disappear rapidly, and thus considering each component of possibly-multiple dynamic factors is very difficult.

4. The subspace identification process in Section B.5 should be, even if very briefly, mentioned somewhere around Eqs. (6)--(8) because it would be a big question when reading there.

5. Just before Eq. (3), $\phi$ should be called as a *left* eigenvector, instead of just saying eigenvector.

**Summary Of The Paper:**

A method for disentangled representation learning from sequential data is proposed. It is based on the eigendecomposition of an estimation of the Koopman operator of dynamics. The method is demonstrated with standard benchmark datasets.

**Summary Of The Review:**

This is a nice paper introducing an interesting idea for static/dynamic disentanglement of sequential data. While the discussion could be deepened in a few aspects, the current paper looks sufficiently good for publication in ICLR.

---

> ### Author Response · Authors · 2022-11-16
> **Response to Reviewer tz1i - Part1**
>
> We would like to thank Reviewer tz1i for identifying the clarity of our writing, the novelty of using Koopman for disentanglement, and our extensive evaluation of the method. In addition, we would like to thank them for their comments and suggestions for improving the paper and deepen the discussion. Below, we address the comments raised by Reviewer tz1i. Given the opportunity, we will be happy to incorporate the modifications listed below into a final revision.
>
>
> > Despite the nice reasoning for static/dynamic disentanglement, multifactor disentanglement is not well supported by theoretical arguments. I do not think it is a fatal flaw, but it is certainly a weakness as one of the main claims of the paper is the capability of multifactor disentanglement.
>
> We agree with the reviewer that multifactor disentanglement is not fully supported from a theoretical viewpoint. A full theoretic justification will be provided if the Koopman eigenvectors were orthogonal, which holds for e.g., measure-preserving systems as the Koopman operator is unitary. During the design of our method, we experimented with penalty constraints requiring a normal approximate Koopman operator (and thus, admitting a unitary eigendecomposition). However, employing such a strong constraint reduces the space of available matrices significantly, yielding inferior results in our preliminary exploration. We believe that there are better approximative ways to promote orthogonality between factors, while avoiding a reduction in model expressiveness. For instance, recent work incorporated mutual information penalties to VAE frameworks (Li \& Mandt 2018) to improve results and theoretical foundation (e.g., Bai et al. 2021). We leave further investigation and study of this direction for future work.
>
> > The definition of $\mathcal{L_{dyn}}$ penalize $|\lambda\|$ regardless of $< \lambda$ . This sounds strange because even if $|\lambda| = 1$, dynamic modes with $<\lambda\neq0$ represent (energy-preserving) oscillations, which would be regarded as dynamic factors. I guess, in the presented experiments, the current definition of $\mathcal{L_{dyn}}$  worked as expected only because such energy-preserving oscillations were not dominant in the datasets.
>
> We are thankful for Reviewer tz1i's interesting observation regarding the dynamic loss definition. Indeed, we believe that in the benchmarks we consider for sequential disentanglement, energy-preserving oscillation features are not dominant. Importantly, the dynamic loss can yield eigenvalues and thus factors which are close to the unit circle, depending on the value of $0 < \epsilon < 1$. Further, our method could be adapted theoretically and practically to datasets where such features are predominant. Suppose we believe the dataset has such energy-preserving oscillations features. In that case, we could allow the dynamic eigenvalues to have a norm of 1 and be far from eigenvalues on the point $1 +0j$ which represents the static factors. It could be implemented by applying an "$\epsilon$-ball" loss around the point $1 + 0j$ which penalizes dynamic eigenvalues for being inside the $\epsilon$-ball (but they could have the value $|\lambda|=1$). We added a discussion to the new revision in Appendix B.3.3 on additional possible definitions of the dynamic loss for datasets with measure-preserving or growing features.
>
> > Multifactor-ness only holds for static factors. While it might be a limit of the proposed method, it is understandable because, in many datasets, dynamic factors correspond to dissipative dynamic modes (with, say, $|\lambda| < 0.9$ ) that disappear rapidly, and thus considering each component of possibly-multiple dynamic factors is very difficult.
>
> We believe multifactor-ness also holds for dynamic factors, even if these factors are dissipative. For instance, say we have a sequence of images with a ball moving in a spiral motion from the boundary of the image towards its center. For this dataset, two dynamic factors define the data. 1. The rotation angle, and 2. The distance from the center. Both factors will disappear rapidly (the speed depends on the data) until the ball settles at the center of the frame. This type of factors can be obtained with our method, and thus, for data with similar characteristics, our method is expected to produce a good disentanglement including multiple dissipative dynamic factors (two in the example above). Moreover, based on our discussion above regarding the dynamic loss, one can potentially extend the dynamic representation to include preserving factors.

---

> > ### Comment · Reviewer_tz1i · 2022-12-08
> > **Thanks for the response**
> >
> > Thank you for the detailed response. My evaluation does not change (this is a good paper, should be accepted).
> >
> > As for the multifactor-ness of dynamical factors, though I agree with what you wrote, my point was rather from a more practical point of view. That is, theoretically different dynamic factors could be considered for different components of motions, computing them in practice would often be challenging because of the difficulty of computing rapidly damping dynamic modes from noisy data. But this does not affect my evaluation of the paper and is just a comment.

---

> ### Author Response · Authors · 2022-11-16
> **Response to Reviewer tz1i - Part2**
>
> > The subspace identification process in Section B.5 should be, even if very briefly, mentioned somewhere around Eqs. (6)--(8) because it would be a big question when reading there.
>
> This is a great idea, thank you. We added a brief description of it into the main paper as suggested.
>
> > Just before Eq. (3), $\phi$  should be called as a left eigenvector, instead of just saying eigenvector.
>
> We modified the text accordingly. Thank you.

---

### Decision · Program_Chairs · 2023-01-20

**Decision:**

Accept: notable-top-25%

**Justification For Why Not Higher Score:**

This paper proposes a new idea that potentially worth further development, but the theoretical & empirical results are not yet at the breakthrough level. Experimental results are still on the standard benchmarks of sequential disentangled learning.

**Justification For Why Not Lower Score:**

Existing disentangled representation learning methods for sequence generative models mostly use mutual-information based approaches. This paper introduces a different and novel angle (Koopman spectral analysis) to tackle this challenge. Worth highlighting and I'm sure researchers in sequential disentanglement would like to read this paper.

**Metareview: Summary, Strengths And Weaknesses:**

This paper proposed a new disentangled representation learning from sequential data, based on the eigendecomposition of an estimation of the Koopman operator of dynamics. Benchmark results demonstrate the advantages of this approach over existing mutual information based approaches.

Reviewers all agree that this paper's approach is novel and the experiments are great in terms of supporting the developed approaches.

Clarity seems to be an issue for some reviewers and I would recommend the authors to clarify more about the ideas behind Koopman spectral analysis. Better arguments (in terms of theoretical results) on how the eigenspectrum relates to different factors in the dynamics would enhance the paper's impact further.

**Note From Pc:**

if the above contains the word "oral" or "spotlight" please see: "oral" presentation means -> notable-top-5% and "spotlight" means -> notable-top-25%. As stated in our emails, we are disassociating presentation type from AC recommendations

**Summary Of Ac-Reviewer Meeting:**

N/A